# Estimating digital product trade through corporate revenue data

Viktor Stojkoski[1,2], Philipp Koch [1,3], Eva Coll[1,4] & César A. Hidalgo[1,5] ✉

Despite global efforts to harmonize international trade statistics, our understanding of digital trade and its implications remains limited. Here, we introduce a method to estimate bilateral exports and imports for dozens of sectors starting from the corporate revenue data of large digital firms. This method allows us to provide estimates for digitally ordered and delivered trade involving digital goods (e.g. video games), productized services (e.g. digital advertising), and digital intermediation fees (e.g. hotel rental), which together we call digital products. We use these estimates to study five key aspects of digital trade. We find that, compared to trade in physical goods, digital product exports are more spatially concentrated, have been growing faster, and can offset trade balance estimates, like the United States trade deficit on physical goods. We also find that countries that have decoupled economic growth from greenhouse gas emissions tend to have larger digital exports and that digital exports contribute positively to the complexity of economies. This method, dataset, and findings provide a new lens to understand the impact of international trade in digital products.

At the 2016 Consumer Electronics Show in Las Vegas, Netflix announced an expansion to 130 countries[1,2]. This expansion is an example of the explosive growth of digital trade, in this case, that of a subscription service designed to deliver digital goods (films and TV series) across international borders.

But what is digital trade? And how do some institutions define it?

Despite its undeniable importance, defining and measuring digital trade is surprisingly challenging[3–6]. The Handbook on Measuring Digital Trade, a flagship publication prepared jointly by the OECD, WTO, UNCTAD, and the IMF[4], defines digital trade as all trade that is digitally ordered and/or delivered. That includes (i) physical trade that is digitally ordered (e.g. purchasing clothes from a foreign online vendor), (ii) trade involving physical services (e.g. using a foreign app to buy a plane ticket), and (iii) trade in digital services that are digitally delivered (e.g. using a foreign file hosting service). The Handbook also "adopts the convention that goods cannot be delivered digitally," a convention that, while in agreement with current statistical

approaches, may sound at odds with important trade agreements. For instance, the United States–Mexico–Canada Agreement uses the term "digital product" for goods such as a "computer program, text, video, image, sound recording, or other product that is digitally encoded [and] can be transmitted electronically" and the Japan-Switzerland bilateral trade agreement uses the term "digital products" in a definition that includes also digital plans and designs[5].

These discrepancies are understandable because the distinction between goods and services is not as clear in the digital economy as it is in the physical economy. For instance, entrepreneurs and investors[7,8] often use the term product to indicate service-like activities that are made product-like and scalable through automation and self-service. In that world, people make a strong distinction between the digital delivery of a traditional service (e.g. a remote software engineering team) and the digital delivery of a productized service, such as email, maps, or payment platforms. Consider the difference between hiring a human illustrator to generate a drawing and generating a drawing

[1]Center for Collective Learning, ANITI, IRIT, Université de Toulouse & CIAS Corvinus University of Budapest, Budapest, Hungary. [2]Faculty of Economics, University Ss. Cyril and Methodius, Skopje, North Macedonia. [3]EcoAustria – Institute for Economic Research, Vienna, Austria. [4]LEREPS, Sciences Po Toulouse, University of Toulouse Capitole, Toulouse, France. [5]Toulouse School of Economics and University of Toulouse Capitole, Toulouse, France. ✉e-mail: hidalgo.cesar@uni-corvinus.hu

using an AI. The latter, but not the former, scales because it has replaced labor with capital to serve multiple customers at a low marginal cost. These productized services, or digital products, are at the core of modern venture capital and include many successful sectors, such as software-as-a-service (e.g. Canva, Photoshop), video streaming (e.g. Netflix, Disney+), and cloud computing (e.g. AWS, Google Cloud). Indeed, recently a critical policy discussion has emerged regarding the classification of such digital products under trade agreements[9–11]. The outcome of these policy discussions could have a profound impact on the sector, potentially subjecting some digital product transactions to tariffs, thereby affecting the statistics capturing the economic contributions of these digital products.

Our work thus focuses not on all forms of digital trade, but on trade involving digital goods, productized services, and digital intermediation fees, which we call digital products.

First, we have pure digital goods, such as downloadable video games and movies. Pure digital goods have product like properties, such as a high fixed cost to produce the first unit and a negligible marginal cost to produce copies (e.g. video game downloads). They also involve the transfer of a digital asset, such as a song, movie, or video game. Next, we have productized digital services, which involve access to a digitally encoded and automated service, such as platforms that sell data for a fee, cloud computing, or self-service digital advertising in maps, social media, or search. These productized services range from subscription models that provide access to digital products (e.g. data, movies) to services running fully online (e.g. digital advertising). Finally, we consider digital transaction fees, but not the physical trade enabled by the platforms collecting these fees. For instance, we consider the fee collected by a booking site for reserving an accommodation, but not the value of the accommodation stay itself (which involves lodging, a physically delivered service).

These explanatory challenges complicate the estimation of digital product trade. Government bodies estimate digital trade using surveys where companies and/or consumers are asked to self-report the products and services they deliver or purchase online[4]. These probabilistic estimates, however, lack the data provenance and granularity of physical trade data. For instance, they do not disaggregate digital trade into fine-grained categories corresponding to those used by digital firms, such as cloud computing, video streaming, or digital advertising. Instead, they use the less granular categories available in extended balance of payment data, such as computer software, other computer services, or other information services.

But why should we study trade in digital products?

Consider trade balances[12–17]. In 2021, the United States experienced a physical product trade deficit of USD 1.1 trillion (USD 1.63 T in exports and USD 2.73 T in imports)[18]. An important part of this deficit is compensated by digital exports since the US is a net exporter of "bits" (digital products) and a net importer of "atoms" (physical products). Digital trade data can help us get a better picture of trade imbalances around the world[19]. In fact, our methodology to estimate digital product trade is able to capture what is technically known as GATS Mode 3 trade: the trade enabled by the commercial presence of a firm in a foreign country. For instance, the "trade" that Netflix, a multinational firm based in the United States, generates through the presence of its foreign subsidiaries (e.g. Netflix's presence in the Netherlands). Mode 3 trade has been traditionally hard to capture in trade statistics.

Now consider economic decoupling: the separation of greenhouse gas emissions from economic growth[20–24]. Measuring digital trade could illuminate the ongoing debate about the carbon footprint of the digital economy[25,26] by helping compare digital and physical sectors. A possible explanation for the decoupling of important economies, such as that of the United Kingdom and the United States, could be the growth of digital sectors that emit less greenhouse gases per unit of GDP[27,28].

Finally, consider international estimates of economic complexity[29–33], which often leverage international trade data to explain international variations in economic growth[29–31,34–42], income inequality[43–45], and greenhouse emissions[27,46–48]. Without data on the trade of digital products, these estimates may be missing key sectors for advanced economies.

In this paper, we contribute to our understanding of trade in digital products by presenting an approach to estimate it starting from corporate revenue data. Our approach combines machine learning methods with data on the corporate revenues of thousands of large online firms to create bilateral estimates of digital trade for over two dozen sectors. Figure 1 explains our definition of digital trade, with the caveat that our work is not built down from this definition, but up from the corporate revenue data of digital firms (e.g. Alphabet, Meta, Amazon, Uber, etc.). The resulting dataset involves yearly estimates (in USD) of trade in digital products (digital goods, productized services, and intermediation fees) for 189 countries, 31 sectors, and all years between 2016 and 2021. We find that trade in digital products is rather large, at least USD 0.95 trillion in 2021, larger than the GDPs of Saudi Arabia (USD 0.87 T) and Switzerland (USD 0.8 T) during that same year. Yet, because our data does not include the digital delivery of traditional services, or digital trade involving small firms, we obtain values that are lower than the ones reported in the Handbook on Measuring Digital Trade (USD 3.7 T for 2021)[4]. We also find trade in digital products to be growing rapidly, at an annualized rate of 24.6% between 2016 and 2021 (from USD 328 B to USD 956 B) compared to the experienced growth rates of 6% and 4% of goods and services, respectively, during the same period (more comparisons are presented in the results section). Furthermore, we find the geography of trade in digital products to differ from that of trade in physical goods, digitally delivered services, and services, with digital product exports concentrated in fewer origins, but imports being distributed more evenly, similar to recent findings on the impact of digitalization on trade[49]. These findings contribute to our understanding of the role of digital trade in sustainable economic development.

## Results
### Estimating trade in digital products
We construct a dataset of trade in digital products by combining ground truth data on the consumption of digital products for 60 countries with corporate revenue data for over 2500 digital firms (see Methods).

Figure 2 presents a schematic of our procedure. We use machine learning and optimal transport techniques to extrapolate this data to a total of 189 countries and 31 sectors (see Supplementary Table 2 for the countries and sectors covered in our dataset).

We focus on companies involved in digital goods, productized services, and intermediation (e.g. marketplace platforms). Digital goods, such as video games and software, include products in a digital format with a marginal cost of production that is negligible or close to zero (e.g., eBooks, Software). Productized services, such as cloud computing and video streaming, leverage digital means to automate (almost always fully) the provision of a service. This makes the economics of productized services similar to those of manufacturing (low marginal cost for each unit and high fixed cost to initiate production). Finally, we consider also fees collected by intermediation platforms, whether these are involved in the purchase of a physically delivered service (e.g. lodging) or of a digital good or service (e.g. a mobile phone app).

We begin by selecting the largest internet companies (these are firms that do the majority of their business online and have revenues of USD 1 B or more) and manually identifying their subsidiaries from publicly available online sources (e.g., financial statements). Next, we use the Orbis database to gather revenue data (in USD). Orbis is one of the largest databases on firm level data with information on 400+

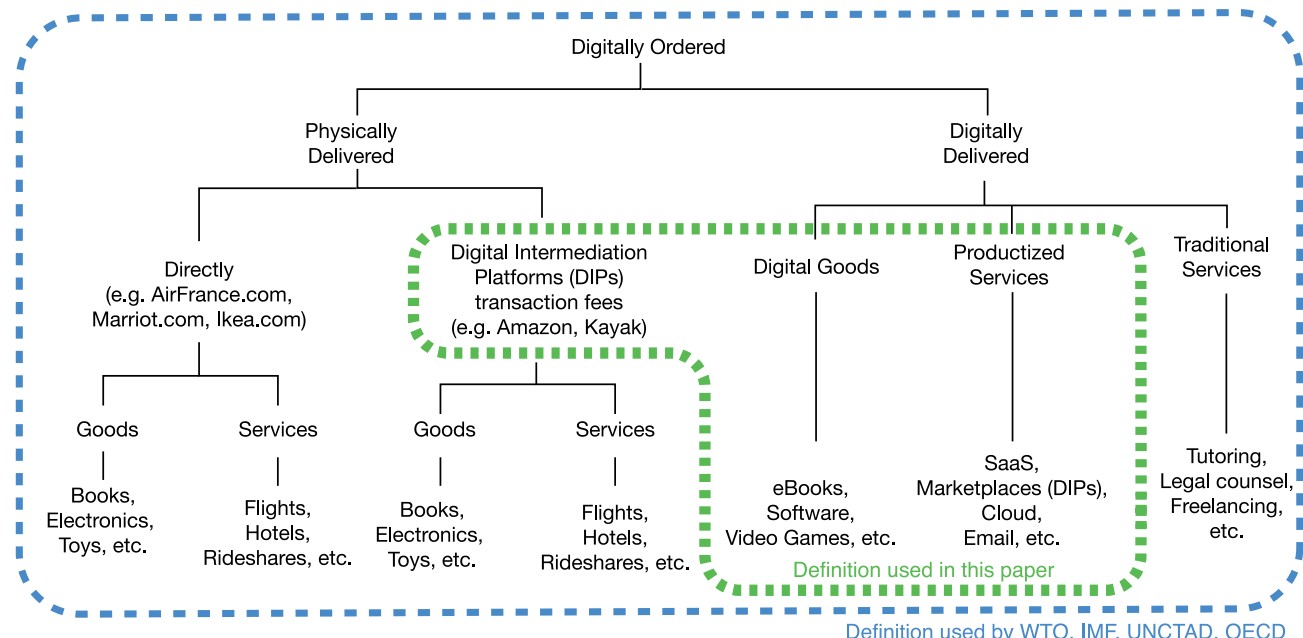

**Fig. 1 | Approximate definition of the bottom-up digital trade data used in this paper.** Digital trade is commonly split among digitally and physically delivered trade. In this paper, we adopt a bottom-up definition starting from data on digital firms that includes digital goods, productized services, and transaction fees in digital intermediation platforms.

million firms across the globe[50,51]. For missing entries, we consult Statista[52], a reliable secondary source for firm revenues. If revenues are still unavailable, we manually collect data from other publicly accessible web sources. We then decompose corporate revenues by digital product sectors using Statista's Digital & Technology Market definitions as a baseline classification. This approach enables us to distinguish 29 digital sectors within the revenue structures of the firms in our dataset.

We then combine the revenue data with country consumption patterns (in USD) from a mobile market intelligence company tracking the consumption of all applications and games downloaded from the Apple's App Store and Google's Play Store for 60 countries (these are two additional digital sectors included in Statista's classification, see Supplementary Table 2 for the countries with available data). We merge these datasets by connecting each firm's sector to its country of origin and to the countries where consumption took place. In Supplementary Note 2 we provide the summary statistics of these data.

In total, we have 31 digital sectors. This enumeration, however, is not exhaustive; and certain digital products, like AI chatbots, are not captured in our analysis.

We use a gradient-boosted regression tree, a flexible supervised machine learning method, to extend the consumption data to an additional set of 129 countries (for a total of 189) and the 29 digital sectors discovered in the firm revenue data. Our model predicts the yearly consumption of digital products within the same sector that belongs to the same parent company for each country. For instance, it estimates the combined consumption in Chile of the cloud computing activities owned by a certain parent company and all of its subsidiaries in 2021. The model's features are motivated by gravity models of trade[53,54] and include parent-category-level variables, such as the total revenues of the parent company in the digital category (across all countries), and the total world consumption of the digital sector (e.g., all app revenues or all games revenues across all countries, see Methods and Supplementary Note 3.1.). We also include features that describe the relationship between the country where the headquarters of the company is located and the country where the product was consumed, such as shared language, borders, common colonizers, the geographic distance between these countries, their respective size in

terms of GDP, and their ICT capacities. We cross-validate our model by using a group-K-fold approach, where we leave 20% (i.e., 5-fold) of the firm-category pairs as a test set, and train the model on the other firm-category pairs. We find that our model has a mean-squared error (MSE) of 23.14, and improves upon a baseline linear regression model (which has a MSE of 24.44). Also, for some of the parent companies included in our analysis we were able to extract the regional consumption from their annual reports (e.g., the total consumption of a multinational company in North America). We used this data to conduct an independent test where we compared the regional shares for the firms with available data with the ones predicted by our model, finding that our model has an MSE of 0.048 (the linear model has a MSE of 0.126, see Supplementary Note 3.2.).

We harmonize the resulting predictions by ensuring that aggregates match their input variables, and by normalizing them to be in accordance with known regional consumption shares. Namely, the cloud computing revenues of a multinational firm across all geographies must equal the total reported cloud computing revenue of that company. Moreover, we normalize our values to match the observed regional consumption shares by assuming that they are the same across different categories (e.g., a multinational firm's revenue share in cloud computing and in digital advertising is the same as the aggregate share).

We allocate the consumption of a firm's digital products to a country of origin using an optimal transport procedure[55,56]. This method assigns consumption to the revenues of the geographically closest subsidiary, without exceeding the subsidiary's revenue. For instance, the combined consumption of all cloud computing activities of a multinational firm in Sweden is first assigned to the cloud revenues of that firm's subsidiaries operating in that sector in Sweden. If these revenues exceed the total consumption of the firm's cloud computing brands in Sweden, then the excess volume is assigned to the geographically closest subsidiary with unallocated revenues. In most cases, we lack information about the revenue share by sector of subsidiaries (we know only those of the parent company) and assume them to be the same as those of the parent company (proportional allocation). In other cases, we are able to manually extract the revenue structure.

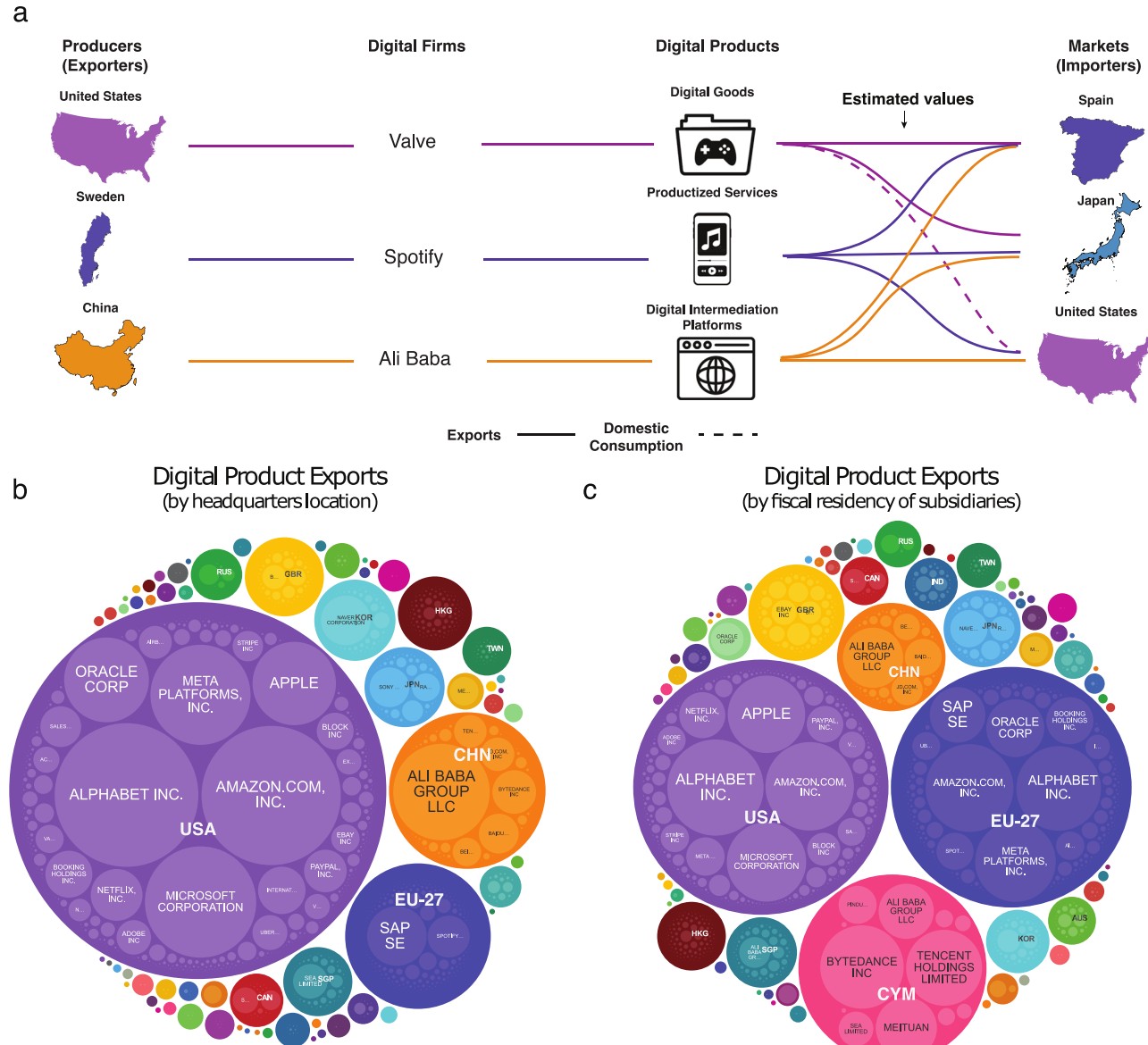

**Fig. 2 | Estimating trade in digital products. a** We estimate bilateral trade in digital products (in USD) for 2,502 firms (belonging to 187 parent companies) and 13,013 important app developers starting from data on their revenue in each digital sector (icons from Envato Elements). We then use a gradient-boosted regression tree to estimate missing digital product consumption links and use optimal transport to assign consumption to firm revenues. **b** Firm level exports when all revenues are assigned to the headquarters location of the company. **c** Firm level exports when the revenues are assigned to the fiscal residence of subsidiaries.

We resort to optimal transport because we do not have information about transactions between parent companies and their subsidiaries or a rule guiding how these transactions take place. Transport methods allocate revenues to consumption by minimizing the distance between export origin and consumption. This leads to conservative estimates prioritizing the allocation of revenues to domestic consumption. To reduce the potential limitation of this assumption, we associate our estimates with upper and lower bounds generated by calculating 95% confidence intervals based on a linear regression that predicts the yearly exports of a firm-category pair and as independent variables uses the revenues of the firm in that category, country origin, and country destination fixed effects.

The international trade of firms operating all the digital sectors covered in our dataset is reported as trade in digitally deliverable services in the Extended Balance of Payments Classification (EBOPS, though there is no one-to-one mapping between the categories, see Supplementary Table 1). Also, the digital products included in our dataset are included in the International Standard Industrial Classification (ISIC) of All Economic Activities (Supplementary Table 1 maps our digital categories into the ISIC classification). Crucially, however, neither the Balance of Payments nor the ISIC distinguish between digital delivery and physical delivery channels.

Our resulting dataset consists of bilateral digital trade estimates for 15,515 firms, 189 countries, and 31 digital product sectors. This dataset, however, does not come free of limitations.

First, the reliance on consumption data primarily from apps and games for forecasting patterns in 29 additional sectors may lead to distortions. This is because the consumption characteristics of these sectors could differ from those observed in app and mobile games data. Furthermore, our assumption that the international trade patterns of digital products align with geographical proximity, as used in our optimal transport allocation, might not always hold true. While this assumption aligns with standard gravity laws of international trade[53],

the minimal physical constraints in digitally delivered trade might break this law.

Another key problem is the assignment of corporate revenues to countries, since digital firms sometimes take legal residence in economies with favorable tax regimes (e.g. Cayman Islands, Luxembourg)[57–60]. In our paper we provide estimates based on two assignment criteria: headquarters location (Fig. 2 b), and the fiscal residence of subsidiaries (Fig. 2 c). Estimates based on subsidiaries may be relevant to those interested in a fiscal view of the data, and unless otherwise noted, are the estimates used in the figures of the main text. In Supplementary Note 4 we also provide estimates assigning all revenues to a company's headquarter, which may be better for those interested in the geography of digital production[61] and those interested in GATS Mode 3 trade. Nevertheless, neither of these assignment criteria are optimal, since not all subsidiaries are legal entities created for tax purposes, and not all product design and development take place in a company's headquarters.

Finally, our estimates are likely to be a lower bound for global trade in digital products because of two reasons. First, our data is based on a limited universe of firms, which is biased towards larger companies (revenues of USD 1B or more). Second, we assign trade to revenues of parents and subsidiaries conservatively, by counting as trade only the digital product consumption that cannot be accounted for by local consumption.

## The growth, geography, and concentration of trade in digital products

We begin by comparing our estimates for trade in digital products with (i) trade in digitally delivered services (DDS) that include our digital sectors plus others (using WTO/UNCTAD data[53]), (ii) trade in services (which should also include our digital sectors), and (iii) trade in physical goods (see Methods for the data used for these comparisons). This helps validate and put in context the estimates we use to understand the growth, geography, and concentration of trade in digital products.

Figure 3a–d compares the aggregate dynamics of trade in physical goods, services, DDS, and digital products between 2016 and 2021. We find that trade in digital products has been increasing rapidly, and that it is comparable to estimates for the dynamics of DDS[4]. Namely, during these years, trade in digital products grew at an annualized growth rate of 24.5% (Fig. 3a), from 320 billion USD in 2016 (95% c.i. lower bound: 275 B, 95% c.i. upper bound: 373 B) to 958 billion USD in 2021 (95% c.i. lower bound: 835 B, 95% c.i. upper bound: 1.10 T). Similarly, trade in DDS grew at an annualized rate of 8% (Fig. 3b), suggesting that digital products play an increasing role in digitally delivered trade. The observed differences between DDS and digital products trade could be a result of the growing productivity of digital products, but also a consequence of the fact that our data is based on the top-performing firms, which are known to experience larger growth rates[62]. In contrast, services (Fig. 3c) and physical goods trade (Fig. 3d) grew moderately, with annualized growth rates of 3.7% and 6.3%. This growth gap accelerated in 2020 during the COVID-19 pandemic, when trade experienced a downturn (trade in physical goods declined by 7%, whereas trade in services declined by 17%), but trade in digital products grew rapidly, year-on-year at a rate of 28% (in the same year trade in DDS grew by only 1%).

For 2021, we estimate trade in digital products to represent around 3.5% of world trade in goods and services (Fig. 3e), making it an area of increasing economic importance. If trade in digital products were to continue to grow at the same annualized rate experienced between 2016 and 2021, we would expect trade in digital products to reach about 15% of global trade by 2030. Figure 3e also shows the estimated composition of trade in digital products compared to trade in services and in physical goods. Trade in digital products is explained mostly by trade in cloud computing, online marketplaces, n.e.s., and digital advertising, which amount to around 65% of all estimated digital

trade (see Supplementary Note 5 for the structure of trade in digital products over the years).

Figure 3f–m compare our estimates of digital product trade with exports (Fig. 2f–h) and imports (Fig. 2i–k) of DDS, services, and physical goods. We observe that the exports of digital products are highly correlated with trade in DDS (Fig. 3f), and that this correlation decreases as we move towards services (Fig. 3g) and physical goods (Fig. 3h). We also observe that imports of digital products are highly correlated with DDS (Fig. 3 i), however, in this case the correlation with services (Fig. 3j) and physical goods (Fig. 3k) does not decrease substantially.

Figure 4 compares the spatial concentration of different forms of trade in 2021 using spike maps (Fig. 4a–h) and Lorenz curves (Fig. 4i, j) of digital products, DDS, services, and physical goods exports and imports. We find that 80% of trade in digital products originates in the top 3% of countries, whereas 80% of digital product imports go to less than 20% percent of countries. Digital product exports (Fig. 4a) are more spatially concentrated than DDS exports (Fig. 4c)[49], service exports (Fig. 4e), and physical exports (Fig. 4g). Digital product exports originate primarily in the United States, but also in, small countries, such as Ireland, Luxembourg, and the Cayman Islands, when we use the assignment rule based on subsidiaries (which favors tax heavens). Digital product imports, however, (Fig. 4b), are not as similar to DDS (Fig. 4d) and service imports (Fig. 4f). Instead, they appear to be less concentrated, with levels of concentration comparable to physical imports (Fig. 4h), suggesting that they are driven by demand factors instead of supply constraints (e.g. knowledge agglomerations[63–65]).

Our results corroborate recent findings about the concentration of digital trade[49]. Nevertheless, trade in digital products encompasses a relatively narrow set of goods or services. Hence, it is still plausible that the observed differences in concentration arise not because of differences in these two forms of trade, but because we expect a smaller set of products to originate in a smaller set of countries. To test this hypothesis, we conduct two robustness checks in the Supplementary Note 6. First, we compare the concentration of trade in digital products with the concentration of trade in each EBOPS services section and in each Harmonized System (HS) goods section (each involving a few dozen products). Second, we perform a simulation where we randomly select physical goods to match the total trade value of the ones available in our dataset of digital products. In both cases, we find that digital products exhibit a substantially higher concentration of exports, indicating that this is not a consequence of simply considering a smaller number of sectors.

The spatial concentration patterns provide, at best, an incomplete picture of the networks of global trade. So next, we compare digital products, DDS, services, and physical goods using network visualizations (Fig. 5a–d). We formalize the position of a country in each of these networks using eigenvector centrality[66–68], a measure of a node's importance in a network. We use the eigenvector rank correlations between the countries in these four networks to study their similarities (Fig. 5e–g, using only the countries with non-zero digital product trade network centrality). The digital products trade network most closely resembles the DDS network, followed by the services network[69]. Indeed, all these networks are centered primarily on the US, with the difference being that in the digital products network tax havens such as Ireland and Luxembourg play a more central role (Fig. 5h). The physical goods trade network[70,71], in contrast, is centered around three regional hubs: The United States, Germany, and China, with China being the most central node in this network.

## Implications of trade in digital products: trade balances, decoupling, and complexity

Having explored the spatial and temporal dynamics of trade in digital products we now turn into their implications. Here, we explore three key implications: trade balances[12–17], the decoupling of greenhouse

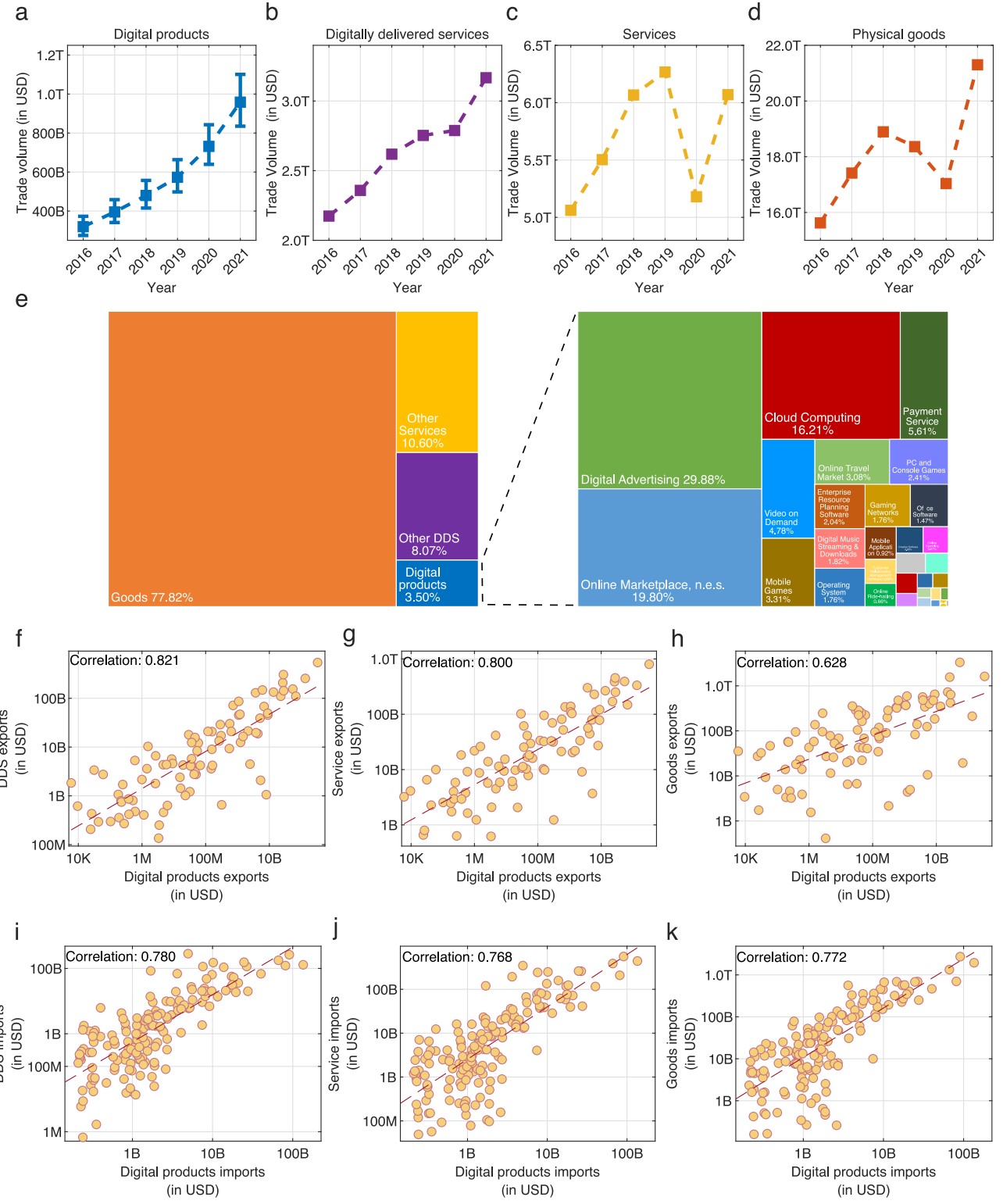

**Fig. 3 | Summary statistics and comparisons of trade in digital products.**
**a** Estimated global trade in digital products in USD (this paper). The error bars show the 95% confidence intervals. **b** Estimated global trade in digitally delivered services in USD (UNCTAD). **c** Global trade in services in USD (UNCTAD) **d** Global trade in physical goods in USD. **e** Estimated composition of trade in digital products compared to services and goods trade in 2021. **f**–**h** Scatter charts comparing countries' exports in 2021 of digital products to the exports in digitally delivered services (**f**), services (**g**), and goods (**h**) networks. **i**–**k** Same as **f**–**h**, just for imports. Figures **f**–**h** use data only for countries with non-zero digital product exports and the presented correlation is between the log values.

gas emissions from economic growth[20–23,72], and estimates of economic complexity[29–33].

First, we use our estimates to understand how digital products trade affects trade balances. Figure 6a, b present comparisons between trade balances in goods and services (x-axis) and trade balance in digital products based on subsidiaries (6a) and headquarters (6b). The latter of these two captures information about GATS Mode 3 trade. In both figures we can clearly observe four

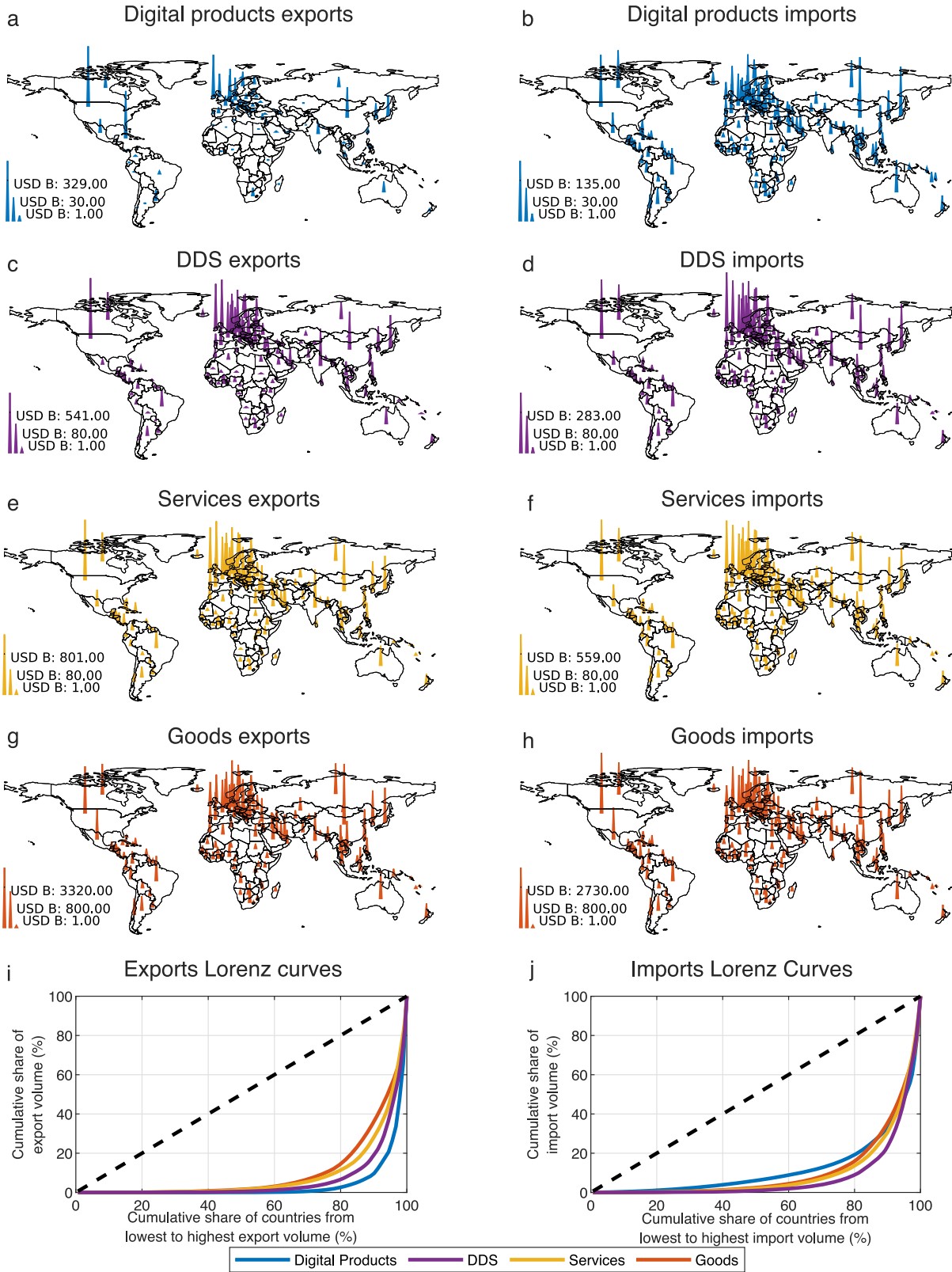

**Fig. 4 | The geography of trade in digital products. a–d** Spike maps showing the spatial concentration of digital products (**a, b**), digitally delivered services (**c, d**), services (**e, f**), and goods exports and imports in 2021 (**g, h**). **i–h** Lorenz curves for the exports (**i**) and imports (**j**) distributions shown in **a–h**.

quadrants. On the top right we have countries with a positive balance of trade in both, goods and services, and in digital products. Using subsidiaries (6a), these are Sweden, Ireland, Luxembourg, and Singapore. Using headquarter assignment (6b), we get Sweden, China, and Singapore, indicating that Ireland and Luxembourg's positive trade balance may be due to them acting as passthrough countries for the GATS Mode 3 trade of other countries, such as Sweden and the United States. On the bottom right, we have economies with a

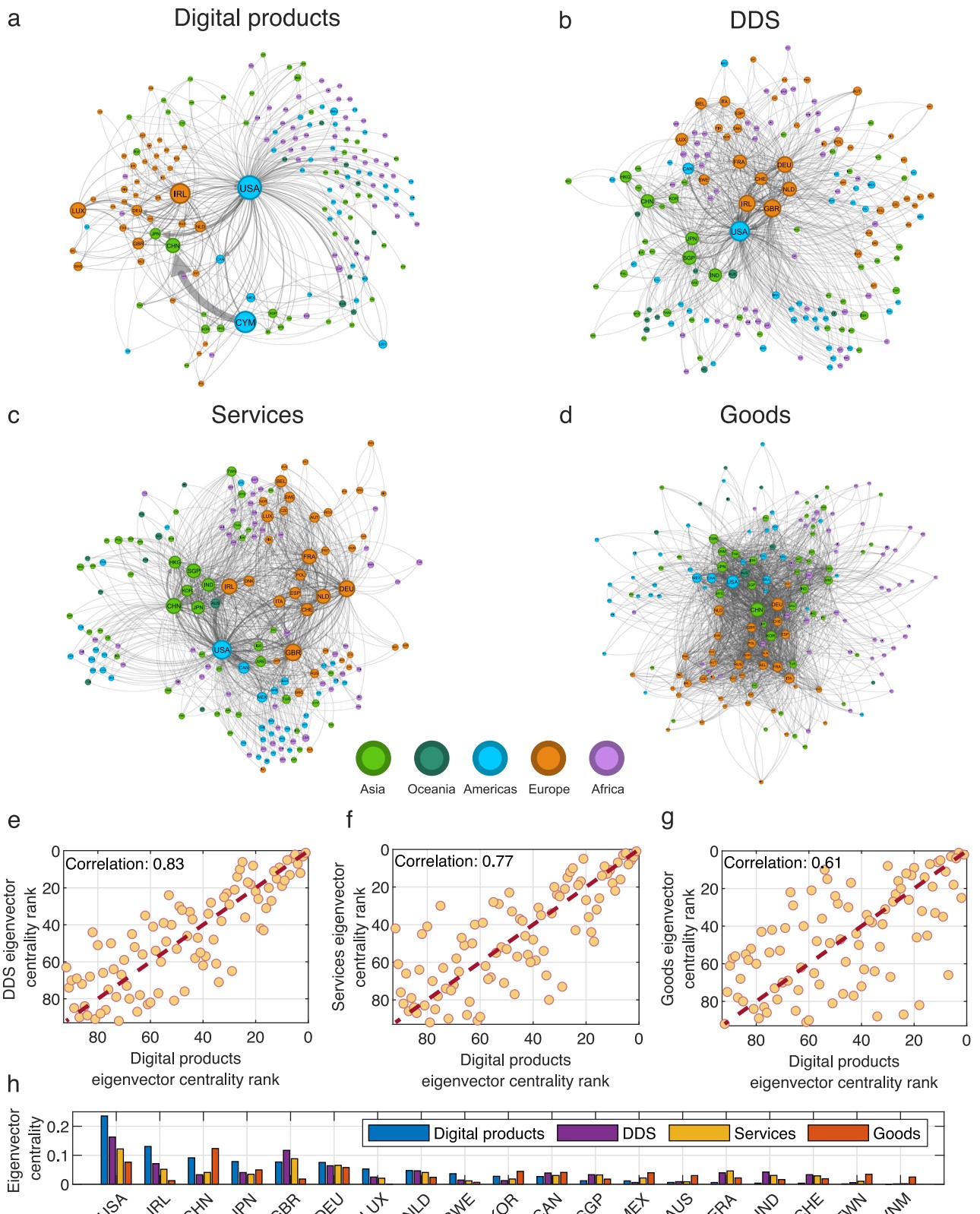

**Fig. 5 | The network structure of digital products trade. a–d** show country-to-country networks of trade in digital products (**a**), digitally delivered services(**c**), and goods (**d**) in 2021. For each country, we show the top import and export destination. We also highlight all bilateral trade flows with a volume above USD 1B. **e–g** Scatter charts comparing the countries' eigenvector centrality rank in the digital products trade network to the centralities in the digitally delivered services (**e**), services (**f**), and goods (**g**) networks. We use data only on countries with non-zero digital product trade eigenvector centrality. **h** Eigenvector centralities for the top 10 countries in terms of eigenvector centralities in all four networks. We exclude countries with only available export data from the eigenvector centrality calculations.

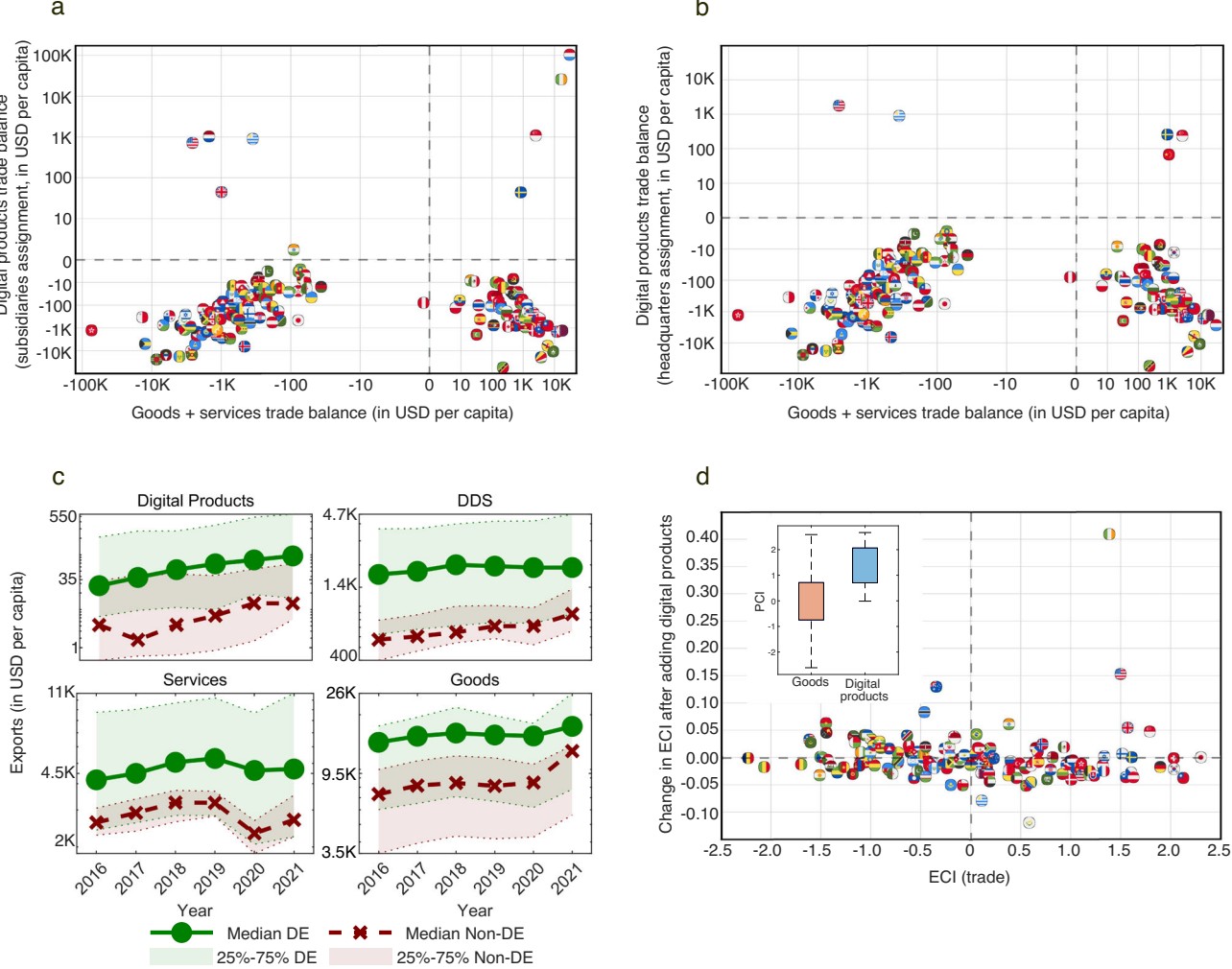

**Fig. 6 | Implications of trade in digital products. a** Total trade balance (goods + services) vs digital product trade balance (USD per capita) in 2021 using the subsidiary assignment for digital products trade. **b** Same as **a**, only using the headquarters assignment for digital products trade. **c** Average digital product, DDS, services, and goods exports per capita between 2016 and 2021 for high income economies depending on whether they decoupled growth from emissions or not (DE – decoupled, Non-DE – not decoupled). We highlight the regions enclosed by the 25th and 75th percentiles. **d** Change in economic complexity index estimates after incorporating trade in digital products to data on physical trade. Inset shows boxplots for the PCI of digital products and physical products in 2021.

trade surplus in goods and services and a trade deficit in digital products. These are natural resource exporters, such as Saudi Arabia, and manufacturing hubs, such as Mexico. The top left quadrant are countries with trade deficits in goods and services and trade surpluses in digital products: the United States, India, Uruguay, the Netherlands, and the United Kingdom, in the case of subsidiary assignment (6a), and the United States and Uruguay when using the headquarters assignment (6b). Finally, the bottom left quadrant is populated by countries with a trade deficit in both, goods and services and digital products. This quadrant is mainly populated by developing economies, such as Cameroon and Paraguay, but also includes some advanced economies, such as Austria. We note that trade in goods and services was anomalous in 2021 due to the COVID-19 pandemic, meaning that countries may have shifted into different quadrants in more recent years.

Next, we explore the correlation between trade in digital products and economic decoupling. This is related to the idea of the twin transition: the notion that economies can transition to lower emissions when digitizing[20–24]. We explore the twin transition by studying the relationship between decoupling of growth and emissions for a restricted sample containing only high-income economies with a population of above 1.5 million in 2021 (decoupling means positive GDP per capita growth and negative emissions per capita change, see Supplementary Note 7.1. for more details about our working definition). We use high-income countries as defined by the World Bank (GDP per capita above USD 13 205), to reduce potential endogeneity issues that may arise since high-income economies are more likely to both, decouple and trade more. In Supplementary Note 7.2., we show results for the full dataset.

We start by dividing countries into those that have and have not decoupled growth from emissions between 2016 and 2019, using production emission estimates from the Global Carbon Budget[73] (in Supplementary Note 7.3. we repeat this exercise using consumption emissions). We then calculate the digital product, DDS, services, and physical exports trends for these two groups (the 25th percentile, median, and the 75th percentile). We find that decoupled high-income economies tend to have larger digital product export sectors compared to non-decoupled economies (Fig. 6c). The 25th percentile of the decoupled economies is of similar size to the median of the non-decoupled. Similar results hold for DDS, whereas for services and goods, we find that the 25th percentile of the decoupled economies is below the median of the non-decoupled. These descriptive results suggest that decoupling emissions from growth might be related with trade in digital products and that digitization and sustainable

development could indeed be a correlated phenomenon (see discussion for possible channels)[24].

Finally, we use our dataset to correct estimates of economic complexity. These are measures of the knowledge intensity of economic structures that are used frequently in economic development studies because of their ability to explain international variations in economic growth, inequality, and emissions[29–32]. The idea is that economies engaged in more sophisticated activities can pay higher wages, produce more output per unit of emissions[27,47], and distribute their income more evenly[74]. While there has been progress in the development of multidimensional approaches to economic complexity[32], as of today, the most widely used metrics rely on physical exports data, and thus, miss key information about digital activities.

We revise estimates of economic complexity by combining our digital product exports estimates by sector with physical export data using the HS4 product categorization (1200+ categories). We focus on goods data rather than DDS or services (which would be a better comparator in practice) because economic complexity calculations require highly disaggregate data which is available for the trade of goods and not for the trade of services. We use this data to estimate the Economic Complexity Index (ECI) and the Product Complexity Index (PCI) of each sector and economy for 2021 (the ECI of a country is the average PCI of its exports. By definition, both ECI and PCI have a mean of 0 and a standard deviation of 1, for more details see Supplementary Note 8.1.)[29–31]. Figure 6d shows that adding digital product exports data reduces the economic complexity estimates of some manufacturing hubs such as Mexico and Slovakia, and increases the complexity of economies involved in the exports of digital products, such as the US, Ireland, and Australia. These changes in complexity are explained by the fact that digital sectors tend to be high in sophistication. The inset of Fig. 6 d compares the PCIs of the 31 digital sectors with that of physical products, showing that digital sectors are−on average−high complexity compared to the ensemble of physical goods. The most complex digital sectors are Digital Advertising and eBooks, whereas the least complex is Online Food Ordering (see Supplementary Fig. 15 for the digital product complexity rankings).

We also test the ability of the ECI corrected for trade in digital products to explain economic growth and emission intensities (GHG per unit of GDP, see Supplementary Note 8.3.). Despite having a relatively short time series data, we find that the digital exports corrected ECI has similar performance at explaining future economic growth (Supplementary Table 4) and emission intensity (Supplementary Table 5) than ECI calculated using only physical trade data.

## Discussion

Trade in digital products has become an essential part of the global economy. Yet, we still know little about its geography, composition, and implications. Here we combined machine learning and optimal transport techniques with data on corporate revenues and consumption of digital products to create bilateral trade estimates for 189 countries and 31 sectors and used them to explore five key facts:

First, we found trade in digital products to be relatively large (almost 3.64% of world trade) and growing rapidly (at a rate of 24% a year).

Second, we found trade in digital products follows a different geography and network structure than other forms trade, being more concentrated in its production and more dispersed in its consumption when compared to trade of all digitally delivered services, all services, and all physical goods.

Third, we found that while trade in digital products represents a relatively small fraction of the global economy, it can impact estimates of trade balances for net digital product exporters and importers.

Fourth, we provided descriptive statistics suggesting that decoupled economies tend to export more digital products, which could mean that digitalization and sustainability are interlinked, as suggested by the twin transition hypothesis[24].

Fifth, and finally, we showed that digital sectors could improve metrics of economic complexity[31,32], revising upwards the complexity estimates of digital exporters such as Ireland, Australia, and the United States.

Our results are compatible with those obtained from other estimates of digital trade[4,49]. But because we use a narrower, bottom-up definition based on firm revenue data, we obtain lower estimates for the total volume of digital trade. This approach, however, has the benefit of allowing us to disaggregate bilateral trade flows into 31 sectors (compared to a dozen EBOPS categories) and follow sectoral definitions that resemble more closely the structure of the industry (e.g. digital advertising, video streaming, cloud computing, instead of other computer services). Moreover, our estimates can disentangle the trade structure of a parent firm and its subsidiaries. This should facilitate tracking statistics that are currently less visible in national accounts (e.g., bilateral trade in Cloud Computing, or evaluating GATS Mode 3: Commercial presence abroad of the General Agreement on Trade in Services[54]), and thus provide a basis for a more detailed data-driven investigations on the implications of digital trade.

But our dataset is also subject to important limitations.

First, our data is not fully comprehensive. Our estimates are limited to a set of large companies and the most traded digital sectors. They do not include direct transactions among private individuals (e.g. a programmer in India selling an app development service to a client in the UK), and do not cover all digital sectors (e.g., AI Chatbots, AI image generation). Also, focusing on the largest companies might lead to overestimating the growth and concentration of digital exports, just because these are high growth frontier firms[62].

Second, our estimates are based on several assumptions. One of them was the use of bilateral data on two digital sectors (mobile apps and games) to train our model and to extrapolate our estimates to 29 additional categories. This data limitation might distort the trade patterns of sectors that have different consumption patterns to mobile apps and games, which are more consumer oriented instead of business-to-business oriented sectors (such as cloud computing). In the future, it may be possible to overcome these limitations with the availability of similar bilateral data for other sectors. We also assumed as little trade as possible by maximizing observable domestic consumption. This leads to conservative estimates that can provide only a lower bound for digital product trade volumes. Lastly, we assumed an optimal transport allocation where trade is assigned to the geographically closest subsidiary. This might not be entirely realistic since digital trade does not involve physical transaction costs.

Third, we are unable to track transactions between parent companies and their subsidiaries. To accommodate this limitation, we provide estimates based on two assignment rules: based on the location of each subsidiary or of the headquarters. Both assignment rules, however, are not ideal, since the location of subsidiaries respond to both, local knowledge pools and tax incentives[75].

Fourth, our data is limited to only six years. This restricts the development of longitudinal studies investigating the long run impact and implications of trade in digital products.

Finally, this dataset alone is not enough to fully explore the role of digitalization on the sustainable economy, as multiple channels could be explaining the observation that decoupling economies also export more digital products. In particular, the effect of digitalization on emissions could occur when a country substitutes a more polluting physical production process for a less polluting digital version. For example, a DVD that is now downloaded could reduce carbon compared to a DVD shipped overseas. The impact of new services, such as cloud computing and data centers, while adding to overall emissions, can reduce emission intensities (emissions per unit of GDP) if the new activity produces more GDP per unit of emission than the average

activity in that economy. These activities, in the case of cloud, web hosting, or video conferencing, could also provide infrastructure—even when they are run by foreign firms—that reduce the emission of other sectors in the economy. For instance, a digital accounting service that decreases the number of physical meetings between an accounting firm and their clients could cut the number of physical trips and their associated emissions. Properly considering the environmental impact of activities such as data center and cloud computing[25,26], requires further research that incorporate indirect effects and comparisons with other sectors of the economy.

Yet, despite these limitations the dataset, method, definitions, and results in this paper advance our understanding of digital trade, widening the door to study one of the key aspects of our global economy.

## Methods

To create our estimates for bilateral trade we collect data on corporate revenues, country consumption patterns (both in USD), and couple them with machine learning and optimal transport methods. With the corporate revenues we identify the origin countries of digital production and the volume of production per country, whereas with the country consumption patterns we discover the consumption volume per country and the consumption destinations. Machine learning helps us augment missing country and digital product data. Finally, transport methods aid us in optimally allocating the revenues to consumption patterns, thereby minimizing the distance between export origins and consumption destinations.

### Revenue data

We begin by identifying the largest internet companies (characterized by annual revenues exceeding USD 1 B in 2020 and a business model predominantly based online), including important app and game developers. Our search for such companies yielded 187 results. Utilizing this data, we then identified important subsidiaries of these major internet firms from publicly available online sources, such as web searches and financial statements. These subsidiaries, often based in different countries, engage in similar business activities as their parent companies. By combining the parent companies and the subsidiaries, we identified a total number of 2502 firms involved in digital production.

We primarily rely on the Orbis database to collect the revenue of these firms. In cases of missing data, we turn to Statista as a secondary source. If necessary, additional revenue information is manually gathered from other publicly accessible web sources (e.g., macrotrends.net, stockanalysis.com, and annual reports).

Since parent companies often report consolidated financial statements, we deduct the revenue of their subsidiaries from the reported value (we do not do this if the revenue reported by all subsidiaries exceeds the parent company revenue).

Orbis is instrumental in identifying the revenues of companies selling a single digital product. However, many firms generate revenue from multiple streams. For such companies, Orbis cannot segregate these streams. This limitation is addressed by using Statista, which provides detailed information on the revenue structure of most parent companies. We apply this data to manually assign digital product sectors to these firms and distribute their revenue accordingly. When Statista lacks specific revenue structure data, we resort to analyzing the companies' financial statements. We generally assume that subsidiaries mirror their parent companies' revenue structures, but when more specific information is available, we apply it to the respective subsidiaries. Through Statista, we determined that our dataset encompasses firms across 29 digital product sectors. Detailed descriptions of these sectors are available in Supplementary Note 1.

Due to constraints in the consumption patterns data (described in the following paragraph), we only collect firm revenues for the period between 2016 and 2021.

### Consumption data

We collect consumption data from a market intelligence company called AppMagic[76]. This data includes detailed consumption patterns for two digital sectors (mobile games and applications) across 60 countries for the years between 2016 and 2021. The data cover consumption patterns on applications and mobile games downloaded from the Apple App Store and Google Play Store, representing most of the mobile application market. These digital products are distributed across 13,013 unique firms and app developers, thus our sectoral coverage to 31 (see Supplementary Note 2 for the yearly summary statistics of this data).

We point out that for certain firms we do not have data on their country of origin. For these firms, we set the origin as the country where the majority of their revenues were generated.

### Machine learning

We use machine learning to augment the consumption patterns data to include the revenues of the largest internet companies across additional 29 digital sectors and to extend the coverage to 129 other countries (we restrict our analysis to countries with available features data – note that not all countries have feature data for all of the years). We do this by using the available consumption patterns data and training a gradient-boosted regression tree to predict a country's yearly consumption in all digital product brands of a target firm that are in the same digital sector (e.g., we separately predict a firm's Advertising revenues and Cloud Computing revenues in Chile). That is, our model predicts the consumption of a firm-digital sector pair in a country.

### Model features

Our choice of input features for the model is motivated by the gravity model of trade. The idea behind this model is that flows should be larger between economies with larger size (in terms of GDP, export volume, or population) and which are geographically close. Also, the volume of this flow should be dependent on common cultural, historic, and economic factors. Here, we make the same analogy and assume that a country's consumption of all digital product brands of a company that are in the same digital sector is dependent on the destination's total consumption and features that describe the relationship between the headquarters' country of origin and the destination country.

We generate the input features of our model by collecting data from several sources (See Supplementary Note 3.1. for definitions and data sources). First, we include digital revenue data taken from the same sources (Orbis, Statista, and AppMagic) that were described in the Revenue data and Consumption data paragraphs. We use this data to create three input features describing the digital size of the exporting company, the country of origin, and the digital sector: 1) the total revenue of all digital products that are in the same sector and under that are headquartered in the same company across the world (in USD), 2) the total revenue of the companies with headquarters coming from the same country of origin across all digital sectors (in USD), and 3) the total world consumption of all products belonging to the same sector (in USD).

Second, we collect data on features that describe the relations between the country where the headquarters of the firm are located and the country that is the destination of the consumption. These are 11 features capturing factors such as common official/unofficial language, colonial and political relationships, geographical proximity, and the regional location of the countries. We

collect these data from Centre d'Etudes Prospectives et d'Informations Internationales.

Third, we collect data on the GDP (in current USD) of the country where the headquarters of the firm are located and the country destination of consumption, and generate two input features. These features help us capture the potential market size effect of the initial market of the product and the destination market. The data for these features are taken from the World Bank's World Development Indicators.

Fourth, we include features that describe the ICT potential of both the country of origin of the headquarters and the destination country and generate six features. For this, we collect data on the share of the population having a fixed broadband connection (2 features), the share of population having mobile broadband connection (2 features), and the share of population using the internet (2 features) from the International Telecommunications Union.

In addition, because our data is zero inflated, we use the train data to estimate a logistic regression for the probability of a non-zero consumption of the product in a country, and use these estimates as an additional input feature in our model. Finally, because we predict values for different years, we also use year dummies as additional features (in the linear regression model this would correspond to period fixed effects).

All input features, except the cultural and historical indicators, are transformed to their logarithmic values. Moreover, to control for the zero values, we add 1 to each observation and feature.

## Model cross-validation

To train the model, we use only the firm-digital sector pairs with yearly revenues above USD 10 million. Removing products with total revenues less than USD 10 million helps us prioritize the information provided by the products that are more widely traded and have a stronger influence on the overall market dynamics. This prevents the model from being overly influenced by the consumption patterns of niche or less representative products.

We tune the hyperparameters of the model by using a group-K-fold cross-validation, where we leave 20% of the firm-category pairs as a test set, and train the model on the other firm-category pairs. We use a grid search over several possible values for maximum tree depth (1,3, 5, 7) and the minimum child weight (5, 20, 50, 100, 200), and fix the learn rate to 0.1 and the number of learning cycles to 150. The idea behind this cross-validation approach is that our main goal is to extend the consumption data to new firm-category pairs.

The average mean-squared-error (MSE) of our model is 23.14. This improves upon a baseline linear regression model which has a limited predictive power (an MSE of 24.44). In Supplementary Note 3.2. we provide more details about the cross-validation of the model and provide additional results for the model performance.

## Post-processing

We map the feature vectors to estimates for the log of digital product consumption in a country and in a given year. We then pass the feature vector for each country-firm-digital-sector pair to produce an estimate for log of the consumption pattern for a given year. We exponentiate these estimates to recover USD values, To reduce noise, we treat every estimate of less than USD 1000 as 0. We harmonize the data by ensuring that aggregates match their input total firm-category revenues (e.g., the total predicted consumption of all products of a firm in Cloud Computing must match their cloud computing revenues). Also, for some of the parent companies included in our analysis we were able to extract the regional consumption from their 2021 annual reports (e.g., the total consumption of a multinational firm in North America). We used this data to provide an additional normalization of our estimates to match the regional consumption shares by assuming that they are the same

across different categories. For example, if a firm offers products in two categories: cloud computing and digital advertising, we assume its revenue share in each of the categories in North America matches its overall share of product consumption in that region.

## Optimal transport

We use optimal transport to match the corporate revenues to the estimated consumption patterns. We do this by assuming that a country's consumption of a digital product is always assigned to the subsidiary (or parent) company that is nearest geographically and whose revenues are lower than the consumption. If the consumption is larger than the revenues of the subsidiary, then the excess volume is assigned to second geographically closest subsidiary that has revenues that are not assigned yet to another country.

Formally, let $R_{op}$ be the revenue (in USD) of firm-digital sector pair $p$ that originates from country $o$. Also, let $C_{dp}$ be the estimated consumption (in USD) of the same firm-digital sector $p$ in destination country $d$. Using optimal transport, we can find the matrix $\boldsymbol{X_p} = [X_{odp}]$, describing the revenue of $p$ that is a result of consumption in $d$ and is distributed to the revenues of $o$ can be found as the solution that maximizes the following linear cost problem

$$\boldsymbol{X_p} = \arg\max \sum_{o,p} W_{od} X_{odp}, \tag{1}$$

subject to,

$$C_{dp} = \sum_o X_{odp}, \tag{2}$$

and,

$$R_{op} = \sum_d X_{odp}, \tag{3}$$

for all $o$ and $d$. In the cost function, $W_{od}$ are cost weights that are inversely proportional to the geographical distance $D_{od}$ between $o$ and $d$, i.e., $W_{od} \sim \frac{1}{D_{od}}$. The constraints ensure normalization of the marginal distributions to the respective country revenue and consumption volumes.

Under this setup, we provide conservative estimates about the international trade of digital products that prioritize allocating revenues to domestic consumption.

## Confidence intervals

For each estimated non-zero revenue $X_{odp}$ of a firm-digital sector pair $p$ that originates from country $o$ and comes from destination $d$, we create 95% confidence intervals for the upper and lower bounds for our estimates. We do this by estimating a linear regression model for each year separately, where the dependent variable is the log of the revenue (in USD) of the firm-digital sector pair in the destination country. We keep the model simple, and as explanatory variables we use only the total revenues of the firm-digital sector pair during the same year, and country origin and destination fixed effects. We estimate the model in two steps. In the first step, we use origin fixed effects for each potential origin country. Then, to reduce the standard error, we group each origin country with a statistically insignificant coefficient into a single category, and estimate the new model. We calculate the 95% confidence interval for the predicted values and normalize them so that the mean estimate matches the estimated revenue (in USD) of the firm-digital sector pair in the destination country.

**Data for comparison with DDS, services, and goods trade**

We use the UNCTAD/WTO services dataset to compare our estimates with known results on digitally delivered services and services. This dataset provides country-level exports/imports for 12 different broad services categories. Six of these represent digitally deliverable services (but not necessarily digitally delivered): SF: Insurance and pension services, SG: Financial services, SH: Charges for the use of intellectual property n.i.e., SI: Telecommunications, computer and information services, SJ: Other business services, and SK: Personal, cultural, and recreational services. In our analysis, we exclude the trade volume of SF: Insurance and pension services category since none of the digital sectors that we consider belongs to this category. We use the weights provided by Eurostat to map the digitally deliverable trade volume to the digitally delivered trade volume[77].

Since the UNCTAD/WTO dataset does not cover bilateral trade in great detail (because bilateral data is scarce), for the comparison of bilateral trade networks, we use WTO's experimental BATIS dataset, which provides bilateral trade estimates for the 12 EBOPS categories[53].

The physical goods trade data used throughout the paper was taken from the Observatory of Economic Complexity.

## Data availability

The country level data generated in this study have been deposited in the FigShare database under accession code https://doi.org/10.6084/m9.figshare.26048266.v1. The raw firm level data are protected and are not available due to data privacy laws.

## Code availability

The code needed to reproduce the results is available on Zenodo: https://doi.org/10.5281/zenodo.11855687[78].

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

## Acknowledgements

We acknowledge the support of the Agence Nationale de la Recherche grant number ANR-19-P3IA-0004, the 101086712-Learn-Data-HORIZON-WIDERA-2022-TALENTS-01 financed by European Research Executive Agency (REA) (https://cordis.europa.eu/project/id/101086712), IAST funding from the French National Research Agency (ANR) under grant ANR-17-EURE-0010 (Investissements d'Avenir program), the European Lighthouse of AI for Sustainability [grant number 101120237-HOR-IZON-CL4-2022-HUMAN-02], the Obs4SeaClim 101136548-HORIZON-CL6-2023-CLIMATE-01 and ANITI ANR-19-P3IA-0004.

## Author contributions

V.S.: conceptualization, methodology, software, data curation, validation, formal analysis, investigation, and writing. P.K.: conceptualization, methodology, formal analysis, and writing. E.C.: conceptualization, methodology, formal analysis, and writing. C.A.H.: conceptualization, methodology, formal analysis, writing, and supervision.

## Funding

## Competing interests

C.A.H. is a founder and creator of Datawheel and the OEC (oec.world). V.S., P.K., and E.C. have no competing interests.
