## [Peer Review File · Nature Communications]

Estimating Digital Product Trade through Corporate Revenue DataREVIEWER COMMENTS

Reviewer #1 (Remarks to the Author):

The paper is fascinating and quite relevant given that there are no stats available on the actual trade that is digitally delivered. I can't comment on the model as I am not a quantitative researcher, but I have some recommendations that I think could help to make the paper more relevant for the trade policy and research community:

- Clarify the coverage of trade in digital "products". The word product is usually used in trade to refer to software, e-book and similar things for which there is no agreement on whether they fall under the category of goods or services. So I think that using this word can be misleading. In your analysis you seem to refer to what is generally defined as digital services, so I'd use this terminology instead. You clarify in the annex the list of services included, but the paper would benefit from additional information about why and how these services are selected in the first place. In doing so, you might want to refer to the handbook on measuring digital trade (https://www.oecd-ilibrary.org/trade/handbook-on-measuring-digital-trade-second-edition_ac99e6d3-en) that represents a reference in the trade community to define digital services.
- You use information related to purchases from applications. This appears to me as an important limitation and it should be clarified how this impacts the types of services which are included in the analysis. For example, it is very unlikely that a user would pay for cloud computing or web hosting through an app.
- Part 5 of the Annex in which you compare your data with the official stats on digitally-deliverable services is quite relevant given that you are offering an alternative to what is currently used for research in this area. So I'd suggest moving a part of it to the main text, clarifying the value-added of your new alternative model.
- General comment: I feel the paper is very dense. I understand you want to provide an application of the data, but offering an alternative way of measuring digital services is already per se a huge contribution and the paper would benefit from having more details on how you do that and how your results compare with the existing stats in this area. By adding the second part on complexity, decoupling and the rest, you take away important space to clarify your methodology. I will write this to the editor and recommend focussing on clarifying your methodology, definitions, results of the model, limitations and value-added compared with alternative measures of digital services.

Reviewer #2 (Remarks to the Author):

The paper seeks to better measure trade in digital products with a view to helping us better understand its implications. The paper finds that trade in digital products has grown faster than physical trade and that it is more spatially concentrated. The paper also suggests this has a number of implications, including for trade deficits, green house gas emissions and economic complexity measures.

While there is little doubt that the methodology used is innovative and that the topic is relevant, the analysis could do more to locate the papers' contribution in the existing measurement literature. For example, one of the main findings from the IMF, OECD, UNCTAD, WTO Handbook for Measuring Digital trade (IMF, et al. 2023), which is now in its second revision, is that existing trade statistics, with the exception of de minimis trade, do not systematically underestimate how much digital trade, including digitally delivered trade, is taking place. There is certainly a problem of visibility, that is, trade statistics do not tell us if a book that has been traded has been digitally ordered or not, or whether a particular service has been digitally delivered (both defining features of digital trade as per the statistical definition in the Handbook). The Handbook also notes challenges in attributing the origin and destination of digitally delivered services due to digital transformation enabling delivery from different locations and from companies choosing legal residence in countries with lower taxes (similar to the challenges noted in the paper). This to say that, while the measurement of digital products in

this paper is novel, it does not seem to contribute to a more accurate measurement exercise than what we currently have. The new data also seem to re-create some of the flaws of the existing trade statistics albeit for a smaller category of digital products.

Indeed, the estimates presented in the paper suggest that digital products represent 5% of global trade. However, existing estimates of digitally delivered services trade suggest this represents 13% of global trade in 2022 – that is USD 3.82 trillion (WTO, 2023). Proxy measures suggest that, when incorporating potentially digitally ordered trade this number might rise to 24% (López-González et al. 2023). Moreover, the growth rates of 20% contrast with recent WTO figures which suggest that the growth rate of digitally deliverable services is around “8.1% annually, outpacing goods (5.6 per cent) and other services exports (4.2 per cent) (WTO, 2023). The underestimation in the volume of trade likely reflects the fact that the measure tracks a limited number of products based on multinational activities of firms operating in high-tech sectors (per Table S1). The difference in growth rates is likely due to the fact that the digital product measure is capturing frontier firms (Andrews et al, 2016).

While alignment between statistical definitions and the definition of digital products in the paper is not strictly necessary, and there are certainly reasons for tracking different aspects of this environment, a better understanding of what the measures of digital products are capturing that is not being captured in the trade statistics is warranted. So too is a discussion of what is not being captured in the measures of digital products that does appear in the trade statistics. Indeed, more could also be done to bring some of the insights from the comparison with UNCTAD data to the main text to give the reader a better sense of differences.

All of this is also important in the context of discussions on trade deficits based on physical trade (which cut across discussions of using services trade statistics to get a full sense of the extent of these). The net exports of bits would be covered in the services accounts so the measurement of digital products should not affect trade balances. Unless services are not being included in the measurement of balances or a claim is being made that the measure of digital products is capturing something that the trade statistics are missing (which could be the case but is unclear from the paper). Moreover, the measure of digital products might already incorporate trade being captured in the goods accounts. Understanding how your measure relates to existing trade statistics is therefore key to uncovering issues around trade deficits.

On the debate about digital product sectors emitting less greenhouse gas emissions per unit of GDP, the analysis could do more to describe different channels of transmission. It would seem important to differentiate between a country moving from a more polluting physical version of a digital product to a less polluting digital version, which would reduce emissions against a country that only produces digital products where there was never a physical version. The net impact would be positive in terms of carbon. For example, a DVD that is now downloaded would be carbon reducing, but the use of cloud computing, which is “new”, would add to carbon as it is not substituting an activity that was being done previously. Bearing in mind growing evidence of the carbon footprint of data centres, the paper might also want to discuss the fact that data centres might be located in countries different from where headquarters are located. The correlations in the paper (decoupling emissions and trade in digital products) might just be that, correlations, taking place through other channels. Richer countries tend to simultaneously trade more digital products (or consume these) and have preferences for lower emissions.

That said, this is a very interesting and innovative method to try to get at the difficult question of how much digital trade is taking place and what this implies more broadly. The methodology, in my opinion, is the main contribution of the work. Indeed, if one wanted to look at a more disaggregated breakdown of different elements of digital products, one could presumably do so with these methods. One of the areas where trade statistics has less visibility is Mode 3 – commercial presence abroad. The method could be used to better track this type of trade helping complement existing trade statistics.

Overall, I think the paper would benefit from a wider discussion setting your contribution in the context of the existing measurement debates around digital trade, trying to identify what is and what is not captured by the digital product measure proposed. The paper could then provide a more careful analysis of how this relates to trade deficits, emissions and complexity, trying to better spell out the channels of transmission.

Bibliography

Andrews, D., C. Criscuolo and P. Gal (2016), "The Best versus the Rest: The Global Productivity Slowdown, Divergence across Firms and the Role of Public Policy", OECD Productivity Working Papers, No. 5, OECD Publishing, Paris, <https://doi.org/10.1787/63629cc9-en>.

López González, J., S. Sorescu and P. Kaynak (2023), "Of bytes and trade: Quantifying the impact of digitalisation on trade", OECD Trade Policy Papers, No. 273, OECD Publishing, Paris, <https://doi.org/10.1787/11889f2a-en>.

IMF et al. (2023), Handbook on Measuring Digital Trade, Second Edition, OECD Publishing, Paris/International Monetary Fund,/UNCTAD, Geneva 10/WTO, Geneva, <https://doi.org/10.1787/ac99e6d3-en>.
WTO (2023) Global Trade Outlook and Statistics, Geneva: WTO.

Reviewer #3 (Remarks to the Author):

Overall

Generating reliable estimates of digital trade is extremely challenging, and existing methods rely heavily on a combination of national statistics and company surveys (see discussion in IMF et al (2023)). Stojkoski et al. focus on a narrower set of products (those that exist solely or primarily in digital form) than much recent analysis of digital trade. To study trade flows in these 'digital products' they employ novel data sources, namely corporate sources of data on the production and consumption of digital products, and supervised machine learning and optimal transport techniques which are 'motivated by' gravity models.

The paper has the potential to make a significant contribution. However, it suffers from the major shortcoming that it does not dialogue sufficiently robustly with existing studies that seek to define and estimate digital trade. This makes it hard to discern whether, and to what extent, the findings are novel. I discuss this further below and highlight some other aspects of the paper that require attention.

Use of the terms 'digital products' and 'digital trade'

The paper focuses on analysing trade in what the authors term 'digital products' which they define as "the international commerce of goods and services... that exist solely or primarily in digital form" (p2). This focus on 'digital products' is much narrower than 'digital trade' as conventionally defined, which is, in my view, a potential strength of the paper. However, the authors should be explicit at the outset in contrasting their analysis of 'digital products' with conventional definitions of 'digital trade', explaining the rationale for their approach and the value-add that comes from analysing this narrower set of products. They also need to make sure that 'digital trade' and 'trade in digital products' are not used interchangeably in the paper as this can be confusing (eg p8 and Figure 2).

Stojkoski et al also take a radically different approach to the prevailing literature in their conceptualisation of digital products as both goods and services. They distinguish between digital

goods which entail the acquisition of a digital asset (e.g. a consumer downloading a copy of a video game) and digital services where no asset is acquired (e.g. provision of temporary access to cloud computing) (p2). In contrast, WTO et al. (2023) define digital trade as 'all international trade that is digitally ordered and/or delivered', focusing on the modality of trade rather than the product characteristics. Crucially, WTO et al (2023) argue (by convention) that goods cannot be delivered digitally (p.27) and thus all products that are 'inherently digital' (the products that Stojkoski et al focus on) are defined as services (p69).

The challenge with taking this different approach is that it is unclear from the paper as how, in practice, they distinguish between 'digital products' and other goods and services that have digital components. They identify 29 different "digital product and services sectors" (Table S1) and it is not clear whether this is an exhaustive list, how the categories are generated, and how they have decided which products are in and out of scope. The title of Table S1 makes it unclear as to whether, in the eyes of the authors, all 29 are "digital products" or some fall into a different category of "digital services". Why for instance, are 'online food ordering' and 'online dating' stand-alone digital product sectors, but no mention is made of online ride-hailing (e.g. Uber) or online accommodation (e.g. Airbnb)? At face value all of these would seem to be forms of 'online marketplace' (itself a separate category). To what extent does their classification of 'digital products' align with the conventional classifications used in digital trade, including ICT services (ISIC 61,61,63 and related sub-sections) which were updated in 2022 to include many digital products (e.g. cloud computing, search)?

Data and estimation of trade in digital products

Stojkoski et al. take a novel approach to estimating patterns of cross-border trade in digital products, which is welcome and has the potential to offer novel insights. Again, it would be helpful to have a discussion on the relative merits and demerits of their approach compared with that used in other recent studies (e.g. Gonzalez, Sorescu and Kaynak 2023). (Part of the Annex (p23-24) does engage with the UNCTAD 'digitally deliverable services dataset' this is only used as a robustness check).

The authors first estimate the global production of digital products, using corporate revenue data from Orbis and Statista. This occurs in two steps. They use Orbis to identify the 245 largest parent firms (revenues of US\$1bn or more) and their subsidiaries (3,473 firms) that are involved in the trade of digital products and then use Statista data to decompose corporate revenues into 27 different digital product sectors. They also obtain data from a mobile market intelligence company that tracks consumption of applications and games downloaded from Apple's App Store and Google's Play Store. They use this to identify a further 4,812 firms producing digital products in two additional sectors (mobile apps and mobile games). In total, the dataset on production of digital products covers 8,530 firms and 9,446 digital product brands, clustered into 29 distinct digital sectors. The authors are clear on two limits of the dataset: it is biased towards large firms, and it is hard to accurately assign the geographic location of production due to tax optimisation strategies.

To generate an estimate of trade flows, the authors need to obtain country-by-country data on consumption, and here there is very little data. The only data they find is on consumption of mobile apps and games which covers 60 countries. From this data they impute consumption data for a further 69 countries and the remaining 27 product sectors. They predict consumption patterns by introducing a range of country-specific factors (ranging from market size to the level of ICT penetration). Trade estimates are then generated by using an optimal transport approach to assign consumption to the geographically closest company.

The strategy for estimating consumption patterns relies on extrapolating from very few true data points (2 sectors, 60 countries) and while they use impressive methods, the authors should be much more explicit about their assumptions and on the limits of this strategy. What assumptions are being made in extrapolating from consumption patterns in 2 sectors to 29, and how realistic are these assumptions? How plausible are the assumptions underpinning an optimal transport approach in a

digital context? For instance, in the case of cloud computing, digital advertising, and online marketplace (the biggest categories in their dataset) are consumption patterns as strongly shaped by geographic factors as in the physical world (e.g. firms tend procure advertising services via their local Google subsidiary / use the closest AWS subsidiary for cloud computing services)? How have they factored differences in the relative importance of geographic distance into their modelling?

Analysis of Trends

Based on these estimates, the authors seek to establish novel insights into the importance and nature of production and consumption of 'trade in digital products' relative to other trade flows. To do this, the authors compare their estimates of 'trade in digital products' with 'physical trade' (their term). They estimate 'trade in digital products' to have represented about 5% of world trade in 2021 and expect it to grow to 13% of global trade by 2030. 70% of this trade is from three sectors (cloud computing, online marketplace, and digital advertising) (p.8).

The authors do not engage robustly with recent work on digital trade, which makes it hard to discern the value-add of their analysis. For instance, the WTO estimates that in 2022, 'digitally delivered services trade' accounted for 12% of global trade. To what extent is the authors' analysis of 'trade in digital products' already captured conceptually and empirically in existing studies on 'digitally delivered services'? Alternatively put, how does their estimate of 'trade in digital products' add new insights over and above those generated by other studies?

A further problem arises from their omission of services trade when the authors compare 'trade in digital products' with wider trade flows, even though services trade accounts for about a quarter of global trade (c.f. Figure 2 which only compares 'physical trade' with 'digital trade'). One of their findings is that trade imbalances in 'physical trade' are partially offset when 'trade in digital products' is considered, and they cite the example of the US (p. 13). However, this is not a particularly novel insight. It is widely known that while the US has a large trade deficit in goods, this is partially offset by its trade surplus in services. It is widely known that US services trade is increasingly digital, with digitally delivered services accounting for 50% of all US services trade, and 20% of all US exports. The insights into the geographic concentration of 'trade in digital products' are fascinating, but again there is recent analysis of the geographic concentration of digital trade flows, which reaches similar conclusions (e.g. Gonzalez, Sorescu and Kaynak 2023 – see p9-10 Annex A).

References

López González, J., S. Sorescu and P. Kaynak (2023), "Of bytes and trade: Quantifying the impact of digitalisation on trade", OECD Trade Policy Papers, No. 273, OECD Publishing, Paris,<https://doi.org/10.1787/11889f2a-en>.

IMF et al. (2023), Handbook on Measuring Digital Trade, Second Edition, OECD Publishing, Paris/International Monetary Fund,/UNCTAD, Geneva 10/WTO, Geneva,<https://doi.org/10.1787/ac99e6d3-en>.

Response to Reviewers

In the following we provide detailed answers to the reviewers' comments (in *italics and in blue*). We have also highlighted changes to the manuscript.

Reviewer #1:

Reviewer #1 (Remarks to the Author):

Overall Assessment

The paper is fascinating and quite relevant given that there are no stats available on the actual trade that is digitally delivered. I can't comment on the model as I am not a quantitative researcher, but I have some recommendations that I think could help to make the paper more relevant for the trade policy and research community:

Response: We thank the reviewer for their encouraging comments regarding our manuscript and for acknowledging the value of our contribution.

Comment 1

- Clarify the coverage of trade in digital "products". The word product is usually used in trade to refer to software, e-book and similar things for which there is no agreement on whether they fall under the category of goods or services. So I think that using this word can be misleading. In your analysis you seem to refer to what is generally defined as digital services, so I'd use this terminology instead. You clarify in the annex the list of services included, but the paper would benefit from additional information about why and how these services are selected in the first place. In doing so, you might want to refer to the handbook on measuring digital trade (https://www.oecd-ilibrary.org/trade/handbook-on-measuring-digital-trade-second-edition_ac99e6d3-en) that represents a reference in the trade community to define digital services.

Response: We thank the reviewer for this comment and agree on the importance of having a clearer definition of the trade involved. In the new version of the manuscript, we provide a clearer

explanation of the definition used in the introduction including a new figure differentiating our definition with that of presented in the handbook of digital trade. This section in the introduction now reads:

“But what is digital trade? And how do institutions define it?”

Despite its undeniable importance, defining and measuring digital trade is surprisingly challenging.³⁻⁶ The Handbook on Measuring Digital Trade, a flagship publication prepared jointly by the OECD, WTO, UNCTAD, and the IMF⁴, defines digital trade as all trade that is digitally ordered and/or deliverable (but not necessarily digitally delivered). That includes, (i) physical trade that is digitally ordered (e.g. purchasing clothes from a foreign online vendor), (ii) trade involving physical services (e.g. using a foreign app to buy a plane ticket), and (iii) trade in digital services that are digitally delivered (e.g. using a foreign file hosting service). The Handbook also “adopts the convention that goods cannot be delivered digitally,” a convention that is at odds with key trade agreements. For instance, the United States–Mexico–Canada Agreement (USMCA) uses the term “digital product” for goods such as a “computer program, text, video, image, sound recording, or other product that is digitally encoded [and] can be transmitted electronically” and the Japan-Switzerland bilateral trade agreement uses the term “digital products” in a definition that includes also digital plans and designs.⁵

These discrepancies are understandable because the distinction between goods and services is not as clear in the digital economy as it is in the physical economy. For instance, entrepreneurs and investors¹ often use the term product to indicate service-like activities that are made product-like and scalable through automation and self-service. In that world, people make a strong distinction between the digital delivery of a traditional service (e.g. a remote software engineering team, freelance voice recording) and the digital delivery of a productized service, such as email, maps, or payment platforms. Consider the difference between hiring a human illustrator to generate a

¹ For example, Marc Andreessen, co-founder of Netscape and the venture capital firm Andreessen Horowitz uses the word products to describe scalable digital goods.⁷ Paul Graham, co-founder of Y-combinator, one of the most influential startup accelerators in Silicon Valley also uses this terminology⁸.

drawing and generating one using an AI. The latter, but not the former, scales because it has replaced labor with digital capital in a way that it allows it to service multiple customers at low marginal costs. These productized services, or digital products, are at the core of modern venture capital and include many successful sectors, such as software-as-a-service (e.g. Canva, Photoshop), video streaming (e.g. Netflix, Disney+), and cloud computing (e.g. AWS, Google Cloud).

Our work thus focuses not on all forms of digital trade, but on trade involving digital goods, productized services, and digital intermediation fees (Figure 1), which we call digital products. First, we have pure digital goods, such as downloadable video games and movies. Pure digital goods have product like properties, such as a high fixed cost to produce the first copy and a negligible marginal cost to produce additional units (e.g. additional video game downloads). They also involve the transfer of a digital asset, such as a song, movie, or video game. Next, we have productized digital services, which involve access to a digitally encoded and automated service, such as platforms that sell data for a fee, cloud computing, or self-service digital advertising in maps, social media, or search. These productized services range from subscription models that provide access to digital products (e.g. data, movies), to services that run fully online (e.g. advertising on Google or Facebook). Finally, we consider digital transaction fees, but not the physical trade enabled by these platforms. For instance, we consider the fee collected by a travel site selling an airplane ticket, but not the value of the ticket (which involves the flight, a physically delivered service).”

Figure 1. Approximate definition of the bottom-up digital trade data used in this paper. Digital trade is commonly split among digitally and physically delivered trade. In this paper, we adopt a bottom-up definition starting from data on digital firms that includes digital goods, productized services, and transaction fees in digital intermediation platforms

Comment 2

- You use information related to purchases from applications. This appears to me as an important limitation and it should be clarified how this impacts the types of services which are included in the analysis. For example, it is very unlikely that a user would pay for cloud computing or web hosting through an app.

***Response:** We agree with the reviewer’s comment that the reliance on consumption data primarily from apps and games for forecasting patterns in 29 additional sectors may lead to distortions. This could be because the consumption characteristics of these sectors could differ significantly from those observed in app and mobile games data.*

We have expanded the section about limitations in both the “Estimating Digital Trade” and the “Discussion” sections of the paper.

In the “Estimating Digital Trade” section we now say:

“First, the reliance on consumption data primarily from apps and games for forecasting patterns in 29 additional sectors may lead to distortions. This is because the consumption characteristics of these sectors could differ significantly from those observed in app and mobile games data.”

And in the “Discussion” section we discuss this limitation in more detail.

“Second, our estimates are based on several assumptions. One of them was the use of bilateral data on two digital sectors (mobile apps and games) to train our model and to extrapolate our estimates to 29 additional categories. This data limitation might distort the trade patterns of sectors that have significantly different consumption patterns to mobile apps and games which are more consumer oriented, instead of business-to-business oriented sectors (such as cloud computing). In the future, it may be possible to overcome these limitations with the availability of similar bilateral data for other sectors.”

Comment 3

- Part 5 of the Annex in which you compare your data with the official stats on digitally-deliverable services is quite relevant given that you are offering an alternative to what is currently used for research in this area. So I'd suggest moving a part of it to the main text, clarifying the value-added of your new alternative model.

Response: *We agree with the reviewer that comparing our estimates with existing data on digitally delivered services can significantly enhance the value and relevance of our contribution. In the revised manuscript, we have expanded this comparison with WTO/UNCTAD data for total exports/imports of digitally deliverable services as a baseline. We adjust these figures to focus specifically on digitally delivered services, applying the weights provided by EUROSTAT (they are also given in the Handbook for Measuring Digital Trade). This comparison is made with a careful selection of digitally delivered EBOPS sections that correspond to the digital products in*

our dataset(i.e., we exclude the Insurance sector, for more details we refer to the Materials and Methods section).

Throughout the results section, we then utilize these revised estimates to draw parallels between the growth and geographical spread of digital products trade and digital trade more broadly. Our findings reveal that while both digital products and digitally delivered trade have been experiencing growth over the years, digital products demonstrate a faster growth rate (which could be due to the fact that we focus only on the most important firms in the digital realm). Additionally, we observe a strong correlation between the exports, imports, and network structure of digital products with their digitally delivered services counterparts. This comparative analysis not only underscores the dynamics of digital product trade but also positions our model as a valuable alternative and complement to the current frameworks used in digital trade research.

These comparisons are now available in three figures on the main text.

Figure 3. Summary statistics and comparisons of trade in digital products. *a* Estimated global trade in digital products in USD (this paper). The error bars show the 95% confidence intervals. *b* Estimated global trade in digitally delivered services in USD (UNCTAD). *c* Global trade in services in USD (UNCTAD) *d* Global trade in physical goods in USD (OECD.world). *e* Estimated composition of trade in digital products compared to services and goods trade in 2021. *f-h* Scatter charts comparing countries' exports in 2021 of digital products to the exports in digitally delivered services (*f*), services (*g*), and goods (*h*) networks. *i-k* Same as *f-h*, just for imports. Figures *f-h* use data only for countries with non-zero digital product exports.

Figure 4. The geography of trade in digital products. *a-d* Spike maps showing the spatial concentration of digital products (*a-b*), digitally delivered services (*c-d*), services (*e-f*), and goods exports and imports in 2021 (*g-h*). *i-h* Lorenz curves for the exports (*i*) and imports (*j*) distributions shown in *a-h*.

Figure 5. The network structure of digital products trade. *a-d* show country-to-country networks of trade in digital products (**a**), digitally delivered services (**b**), services(**c**), and goods (**d**) in 2021. For each country we show the top import and export destination. We also highlight all bilateral trade flows with a volume above USD 1B. *e-g* Scatter charts comparing the countries' eigenvector centrality rank in the digital products trade network to the centralities in the digitally delivered services (**e**), services (**f**), and goods (**g**) networks. **h** Eigenvector centralities for the top 10 countries in terms of eigenvector centralities in all four networks.

Comment 4

- General comment: I feel the paper is very dense. I understand you want to provide an application of the data, but offering an alternative way of measuring digital services is already per se a huge contribution and the paper would benefit from having more details on how you do that and how your results compare with the existing stats in this area. By adding the second part on complexity, decoupling and the rest, you take away important space to clarify your methodology. I will write this to the editor and recommend focussing on clarifying your methodology, definitions, results of the model, limitations and value-added compared with alternative measures of digital services.

***Response:** We thank the reviewer once again for their constructive feedback. We agree with their assessment regarding the density of the paper and the need to focus more on the methodology and comparative aspects of our research. In the revised manuscript, we have placed greater emphasis on describing our method in detail and elucidating how our dataset compares and contrasts with the existing body of knowledge on digital trade (see figures in previous comment).*

While we acknowledge the importance of a thorough methodology explanation, we also believe that the section on the implications for trade balances, twin transition, and economic complexity is crucial for balancing the depth of our methodological exploration with the broader implications of our findings. These contributions were also valued by the other reviewers. Therefore, we kept a revised and a slightly more restrictive version of them in the manuscript.

Reviewer #2 (Remarks to the Author):

Overall assessment

The paper seeks to better measure trade in digital products with a view to helping us better understand its implications. The paper finds that trade in digital products has grown faster than physical trade and that it is more spatially concentrated. The paper also suggests this has a number of implications, including for trade deficits, green house gas emissions and economic complexity measures.

Response: We thank the reviewer for their thoughtful and thorough review of our manuscript.

Comment 1

While there is little doubt that the methodology used is innovative and that the topic is relevant, the analysis could do more to locate the papers' contribution in the existing measurement literature. For example, one of the main findings from the IMF, OECD, UNCTAD, WTO Handbook for Measuring Digital trade (IMF, et al. 2023), which is now in its second revision, is that existing trade statistics, with the exception of de minimis trade, do not systematically underestimate how much digital trade, including digitally delivered trade, is taking place. There is certainly a problem of visibility, that is, trade statistics do not tell us if a book that has been traded has been digitally ordered or not, or whether a particular service has been digitally delivered (both defining features of digital trade as per the statistical definition in the Handbook). The Handbook also notes challenges in attributing the origin and destination of digitally delivered services due to digital transformation enabling delivery from different locations and from companies choosing legal residence in countries with lower taxes (similar to the challenges noted in the paper). This to say that, while the measurement of digital products in this paper is novel, it does not seem to contribute to a more accurate measurement exercise than what we currently have. The new data also seem to re-create some of the flaws of the existing trade statistics albeit for a smaller category of digital products.

Indeed, the estimates presented in the paper suggest that digital products represent 5% of global trade. However, existing estimates of digitally delivered services trade suggest this represents 13% of global trade in 2022 – that is USD 3.82 trillion (WTO, 2023). Proxy measures suggest that, when incorporating potentially digitally ordered trade this number might rise to 24% (López-González et al. 2023). Moreover, the growth rates of 20% contrast with recent WTO figures which suggest that the growth rate of digitally deliverable services is around “8.1% annually, outpacing goods (5.6 per cent) and other services exports (4.2 per cent) (WTO, 2023). The underestimation in the volume of trade likely reflects the fact that the measure tracks a limited number of products based on multinational activities of firms operating in high-tech sectors (per Table S1). The difference in growth rates is likely due to the fact that the digital product measure is capturing frontier firms (Andrews et al, 2016).

While alignment between statistical definitions and the definition of digital products in the paper is not strictly necessary, and there are certainly reasons for tracking different aspects of this environment, a better understanding of what the measures of digital products are capturing that is not being captured in the trade statistics is warranted. So too is a discussion of what is not being captured in the measures of digital products that does appear in the trade statistics. Indeed, more could also be done to bring some of the insights from the comparison with UNCTAD data to the main text to give the reader a better sense of differences.

***Response:** We thank the reviewer for their insightful comments and agree that our methodology, while innovative, tracks a limited range of digital products that we did not describe clearly. In the revised version, we clarify our definition in the text and include a figure comparing our definition with that used in the handbook for measuring digital trade. We now explain how our definition is different from that of the handbook, since it excludes digitally ordered but physically delivered trade in goods and services (e.g. books from amazon, flights from kayak). We now also dedicate several paragraphs in the introduction to explain the categories of digital trade captured in our bottom-up approach, which we group under the label “digital products.” The introduction now provides the following context.*

“But what is digital trade? And how do institutions define it?”

Despite its undeniable importance, defining and measuring digital trade is surprisingly challenging.^{3–6} The Handbook on Measuring Digital Trade, a flagship publication prepared jointly by the OECD, WTO, UNCTAD, and the IMF⁴, defines digital trade as all trade that is digitally ordered and/or deliverable (but not necessarily digitally delivered). That includes, (i) physical trade that is digitally ordered (e.g. purchasing clothes from a foreign online vendor), (ii) trade involving physical services (e.g. using a foreign app to buy a plane ticket), and (iii) trade in digital services that are digitally delivered (e.g. using a foreign file hosting service). The Handbook also “adopts the convention that goods cannot be delivered digitally,” a convention that is at odds with key trade agreements. For instance, the United States–Mexico–Canada Agreement (USMCA) uses the term “digital product” for goods such as a “computer program, text, video, image, sound recording, or other product that is digitally encoded [and] can be transmitted electronically” and the Japan-Switzerland bilateral trade agreement uses the term “digital products” in a definition that includes also digital plans and designs.⁵

These discrepancies are understandable because the distinction between goods and services is not as clear in the digital economy as it is in the physical economy. For instance, entrepreneurs and investors² often use the term product to indicate service-like activities that are made product-like and scalable through automation and self-service. In that world, people make a strong distinction between the digital delivery of a traditional service (e.g. a remote software engineering team, freelance voice recording) and the digital delivery of a productized service, such as email, maps, or payment platforms. Consider the difference between hiring a human illustrator to generate a drawing and generating one using an AI. The latter, but not the former, scales because it has replaced labor with digital capital in a way that it allows it to service multiple customers at low marginal costs. These productized services, or digital products, are at the core of modern venture capital and include many successful sectors, such as software-as-a-service (e.g. Canva,

² For example, Marc Andreessen, co-founder of Netscape and the venture capital firm Andreessen Horowitz uses the word products to describe scalable digital goods.⁷ Paul Graham, co-founder of Y-combinator, one of the most influential startup accelerators in Silicon Valley also uses this terminology⁸.

Photoshop), video streaming (e.g. Netflix, Disney+), and cloud computing (e.g. AWS, Google Cloud).

Our work thus focuses not on all forms of digital trade, but on trade involving digital goods, productized services, and digital intermediation fees (Figure 1), which we call digital products. First, we have pure digital goods, such as downloadable video games and movies. Pure digital goods have product like properties, such as a high fixed cost to produce the first copy and a negligible marginal cost to produce additional units (e.g. additional video game downloads). They also involve the transfer of a digital asset, such as a song, movie, or video game. Next, we have productized digital services, which involve access to a digitally encoded and automated service, such as platforms that sell data for a fee, cloud computing, or self-service digital advertising in maps, social media, or search. These productized services range from subscription models that provide access to digital products (e.g. data, movies), to services that run fully online (e.g. advertising on Google or Facebook). Finally, we consider digital transaction fees, but not the physical trade enabled by these platforms. For instance, we consider the fee collected by a travel site selling an airplane ticket, but not the value of the ticket (which involves the flight, a physically delivered service).”

Figure 1. Approximate definition of the bottom-up digital trade data used in this paper. Digital trade is commonly split among digitally and physically delivered trade. In this paper, we adopt a bottom-up definition starting from data on digital firms that includes digital goods, productized services, and transaction fees in digital intermediation platforms

Besides differences in definition, our method also provides some small but important advantages. For instance, our bottom-up approach provides a more granular sectoral classification of digital products than data based on digitally delivered services.

In response to your suggestion, we have also expanded the “Estimating Digital Trade” section. There, we provide again a detailed definition of the digital sectors considered.

“We focus on companies involved in digital goods, productized services, and intermediation (e.g. marketplace platforms). Digital goods, such as video games and software, include products in a digital format with a marginal cost of production that is negligible or close to zero (e.g., eBooks, Software). Productized services, such as cloud computing and video streaming, leverage digital means to automate (almost always fully) the provision of a service. This makes the economics of productized services more similar to those of manufacturing (low marginal cost for each unit and high fixed cost to initiate production). Finally, we consider also fees collected by intermediation

platforms, whether these are involved in the purchase of a physically delivered service (e.g. an airplane ticket) or of a digital good or service (e.g. a mobile phone app).”

We also agree on the importance of contextualizing our findings within the broader landscape of digital trade. Hence, throughout the Results section, we have drawn comparisons with WTO and UNCTAD’s estimates for digitally delivered trade (Figures 3, 4 and 5 of the main text). This comparative analysis helps in elucidating the scope and scale of our estimates in relation to existing data, while also highlighting the unique aspects of our methodology. In particular, Figure 3 a-d provide growth comparisons for digital products, digitally delivered services, services, and physical goods.

Figure 3. Summary statistics and comparisons of trade in digital products. *a* Estimated global trade in digital products in USD (this paper). The error bars show the 95% confidence intervals. *b* Estimated global trade in digitally delivered services in USD (UNCTAD). *c* Global trade in services in USD (UNCTAD) *d* Global trade in physical goods in USD (OECD.world). *e* Estimated composition of trade in digital products compared to services and goods trade in 2021. *f-h* Scatter charts comparing countries' exports in 2021 of digital products to the exports in digitally delivered services (*f*), services (*g*), and goods (*h*) networks. *i-k* Same as *f-h*, just for imports. Figures *f-h* use data only for countries with non-zero digital product exports.

We also acknowledge the differences in growth rates as highlighted by the reviewer, and in the discussion section we added that one of the reasons for the possible differences in the observed growth is because we are focusing on frontier firms.

Comment 2

All of this is also important in the context of discussions on trade deficits based on physical trade (which cut across discussions of using services trade statistics to get a full sense of the extent of these). The net exports of bits would be covered in the services accounts so the measurement of digital products should not affect trade balances. Unless services are not being included in the measurement of balances or a claim is being made that the measure of digital products is capturing something that the trade statistics are missing (which could be the case but is unclear from the paper). Moreover, the measure of digital products might already incorporate trade being captured in the goods accounts. Understanding how your measure relates to existing trade statistics is therefore key to uncovering issues around trade deficits.

***Response:** We appreciate the reviewer's insightful comment highlighting the complexity of measuring digital product trade and its implications for trade balances. In our initial manuscript, we focused on comparing physical goods balances with digital product balances, which, as the reviewer rightly points out, does not encapsulate the entirety of a country's trade dynamics (i.e., while digital products are included in the service section, services in general were excluded from the analysis).*

In the revised manuscript, we have expanded our trade balance analysis to provide a more holistic view (see Results Section, subsection Implications of trade in digital products trade). We now compare total trade balances (encompassing both services and goods) with and without the inclusion of digital products (Figure 6 a and b). This approach allows us to assess the impact of digital product trade more accurately on a country's overall trade balance. We present this analysis through plots that compare the total trade balance against the adjusted trade balance, thereby illustrating the impact of digital product trade on national economies.

Figure 6. Implications of trade in digital products. *a* Total trade balance (goods + services) vs digital product trade balance (USD per capita) in 2021. *b* Digital product, DDS, services, and goods trade balances for the top 10 economies in the digital and physical trade networks (in USD) in 2021. *c* Average digital product, DDS, services, and goods exports per capita between 2016 and 2021 for high income economies depending on whether they decoupled growth from emissions or not. The error bars show the 95% confidence intervals. *d* Change in economic complexity index estimates after incorporating trade in digital products to data on physical trade. Inset shows boxplots for the PCI of digital products and physical products in 2021.

Moreover, we improved our trade network comparison (Figure 5) and now include in Figure 6 *b* the most important countries in the trade networks involving digitally delivered services and overall services. As expected, we observe that most economies with digital products trade surplus have positive digitally delivered trade balances and positive services trade balances (since digital products are included within these categories). Interestingly, we also observe some countries with DDS trade deficits experience a surplus in digital product exports (e.g., Ireland). Differences in

trade among these categories can be important. Consider the physical goods trade deficit of the US, which decreases by 24% in 2021 when we include trade in digital products (USD 255B). Similarly, most of the economies in the physical trade network (Figure 5 h) that are central and have a physical trade surplus experience a decrease in their terms of trade when we consider trade in digital products (e.g., Vietnam decreases their trade surplus by 16%).

Comment 3

On the debate about digital product sectors emitting less greenhouse gas emissions per unit of GDP, the analysis could do more to describe different channels of transmission. It would seem important to differentiate between a country moving from a more polluting physical version of a digital product to a less polluting digital version, which would reduce emissions against a country that only produces digital products where there was never a physical version. The net impact would be positive in terms of carbon. For example, a DVD that is now downloaded would be carbon reducing, but the use of cloud computing, which is “new”, would add to carbon as it is not substituting an activity that was being done previously. Bearing in mind growing evidence of the carbon footprint of data centres, the paper might also want to discuss the fact that data centres might be located in countries different from where headquarters are located. The correlations in the paper (decoupling emissions and trade in digital products) might just be that, correlations, taking place through other channels. Richer countries tend to simultaneously trade more digital products (or consume these) and have preferences for lower emissions.

***Response:** We appreciate the reviewer's observation regarding the nuanced relationship between digital product sectors and greenhouse gas emissions. In response, we have taken steps in our revised manuscript to address these complexities more thoroughly.*

We recognize that our initial findings suggest a correlation between the trade in digital products and reduced emissions, and that this correlation can be a result of different channels of transmission. We agree that the shift from physical to digital versions of a product (like DVDs to digital downloads) may indeed reduce carbon emissions, whereas the impact of an entirely new digital service (such as cloud computing) could be much more difficult to disentangle.

Nevertheless, the latter does not necessarily imply an increase in emission intensities, since a new activity (such as a data center) can still produce more GDP per unit of emissions than existing activities, lowering the emission intensity of an economy which is technically an average. So, the increase in carbon emissions of the new activity would not necessarily lead to an increase in the emission intensity of the economy. This is important because decoupling is about emission intensities.

In light of this comment, we have expanded our discussion section to include the reviewer's critical comment on the potential limitations of applying digital product trade data to addressing decoupling of growth from emissions.

“Finally, this dataset alone is not enough to fully explore the role of digitalization on the sustainable economy as multiple channels could be explaining the observation that decoupling economies also export more digital products. In particular, the effect of digitalization on emissions could occur when a country substitutes a more polluting physical production process for a less polluting digital version. For example, a DVD that is now downloaded could reduce carbon compared to a DVD shipped overseas. The impact of new services, such as cloud computing and data centers, while adding to overall emissions, can reduce emission intensities (emissions per unit of GDP) if the new activity produces more GDP per unit of emission than the average activity in that economy. These activities, in the case of cloud, web hosting, or video conferencing, could also provide infrastructure—even when they are run by foreign firms—that reduce the emission of other sectors in the economy. For instance, a digital accounting service that decreases the number of physical meetings between an accounting firm and their clients would cut the number of physical trips and their associated emissions. Properly considering the environmental impact of activities such as data center and cloud computing,^{21,22} requires further research that incorporate indirect effects and comparisons with other sectors of the economy. ”

Comment 4

That said, this is a very interesting and innovative method to try to get at the difficult question of how much digital trade is taking place and what this implies more broadly. The methodology, in my opinion, is the main contribution of the work. Indeed, if one wanted to look at a more disaggregated breakdown of different elements of digital products, one could presumably do so with these methods. One of the areas where trade statistics has less visibility is Mode 3 – commercial presence abroad. The method could be used to better track this type of trade helping complement existing trade statistics.

Response: We are grateful for the reviewer's recognition of the innovative aspects of our methodology and its potential contributions to understanding digital trade.

We agree with the reviewer that our results could be instrumental in tracking digital trade statistics that are currently less visible in national accounts (e.g., disaggregated bilateral trade flow of Cloud Computing, or evaluating Mode 3: Commercial presence abroad of the General Agreement on Trade in Services). In response to the reviewer's suggestion, we have expanded the explanation of our method and enrich the discussion section to highlight the unique applications of our dataset.

Comment 5

Overall, I think the paper would benefit from a wider discussion setting your contribution in the context of the existing measurement debates around digital trade, trying to identify what is and what is not captured by the digital product measure proposed. The paper could then provide a more careful analysis of how this relates to trade deficits, emissions, and complexity, trying to better spell out the channels of transmission.

Response: We sincerely thank the reviewer for their constructive feedback and insightful recommendations. We fully agree that situating our contribution within the broader context of existing measurement debates around digital trade is crucial for the depth and relevance of our work. By implementing the suggested changes, we hope to provide a clearer and more detailed perspective on how our methodology contributes to, and diverges from, existing practices in measuring digital trade.

Literature referenced by the reviewer

Andrews, D., C. Criscuolo and P. Gal (2016), "The Best versus the Rest: The Global Productivity Slowdown, Divergence across Firms and the Role of Public Policy", OECD Productivity Working Papers, No. 5, OECD Publishing, Paris, <https://doi.org/10.1787/63629cc9-en>.

López González, J., S. Sorescu and P. Kaynak (2023), "Of bytes and trade: Quantifying the impact of digitalisation on trade", OECD Trade Policy Papers, No. 273, OECD Publishing, Paris, <https://doi.org/10.1787/11889f2a-en>.

IMF et al. (2023), Handbook on Measuring Digital Trade, Second Edition, OECD Publishing, Paris/International Monetary Fund,/UNCTAD, Geneva 10/WTO, Geneva, <https://doi.org/10.1787/ac99e6d3-en>.
WTO (2023) Global Trade Outlook and Statistics, Geneva: WTO.

Reviewer #3 (Remarks to the Author):

Overall Assessment

Generating reliable estimates of digital trade is extremely challenging, and existing methods rely heavily on a combination of national statistics and company surveys (see discussion in IMF et al (2023)). Stojkoski et al. focus on a narrower set of products (those that exist solely or primarily in digital form) than much recent analysis of digital trade. To study trade flows in these ‘digital products’ they employ novel data sources, namely corporate sources of data on the production and consumption of digital products, and supervised machine learning and optimal transport techniques which are ‘motivated by’ gravity models.

The paper has the potential to make a significant contribution. However, it suffers from the major shortcoming that it does not dialogue sufficiently robustly with existing studies that seek to define and estimate digital trade. This makes it hard to discern whether, and to what extent, the findings

are novel. I discuss this further below and highlight some other aspects of the paper that require attention.

Response: We sincerely thank the reviewer for their insightful assessment and constructive criticism. We agree with the observation that our study, while employing novel data sources and methodologies, could benefit from a more robust dialogue with existing studies in the field of digital trade measurement. In our responses we provide details about the changes we made to the manuscript which now dialogue strongly with the existing literature on estimating digital trade.

Comment 1

Use of the terms ‘digital products’ and ‘digital trade’

The paper focuses on analysing trade in what the authors term ‘digital products’ which they define as “the international commerce of goods and services... that exist solely or primarily in digital form” (p2). This focus on ‘digital products’ is much narrower than ‘digital trade’ as conventionally defined, which is, in my view, a potential strength of the paper. However, the authors should be explicit at the outset in contrasting their analysis of ‘digital products’ with conventional definitions of ‘digital trade’, explaining the rationale for their approach and the value-add that comes from analysing this narrower set of products. They also need to make sure that ‘digital trade’ and ‘trade in digital products’ are not used interchangeably in the paper as this can be confusing (eg p8 and Figure 2).

Stojkoski et al also take a radically different approach to the prevailing literature in their conceptualisation of digital products as both goods and services. They distinguish between digital goods which entail the acquisition of a digital asset (e.g. a consumer downloading a copy of a video game) and digital services where no asset is acquired (e.g. provision of temporary access to cloud computing) (p2). In contrast, WTO et al. (2023) define digital trade as ‘all international trade that is digitally ordered and/or delivered’, focusing on the modality of trade rather than the product characteristics. Crucially, WTO et al (2023) argue (by convention) that goods cannot be delivered

digitally (p.27) and thus all products that are ‘inherently digital’ (the products that Stojkoski et al focus on) are defined as services (p69).

The challenge with taking this different approach is that it is unclear from the paper as how, in practice, they distinguish between ‘digital products’ and other goods and services that have digital components. They identify 29 different “digital product and services sectors” (Table S1) and it is not clear whether this is an exhaustive list, how the categories are generated, and how they have decided which products are in and out of scope. The title of Table S1 makes it unclear as to whether, in the eyes of the authors, all 29 are “digital products” or some fall into a different category of “digital services”. Why for instance, are ‘online food ordering’ and ‘online dating’ stand-alone digital product sectors, but no mention is made of online ride-hailing (e.g. Uber) or online accommodation (e.g. Airbnb)? At face value all of these would seem to be forms of ‘online marketplace’ (itself a separate category). To what extent does their classification of ‘digital products’ align with the conventional classifications used in digital trade, including ICT services (ISIC 61,61,63 and related sub-sections) which were updated in 2022 to include many digital products (e.g. cloud computing, search)?

***Response:** We thank the reviewer for their detailed and insightful comment. We acknowledge and agree with the need for clarity in our use of terms and in distinguishing our approach from conventional definitions of digital trade.*

In response, we have revised the introduction and the “Estimating trade in digital products” sections to include a formal definition on how we build our digital trade dataset, as was also suggested by Reviewers 1 and 2.

“But what is digital trade? And how do institutions define it?”

Despite its undeniable importance, defining and measuring digital trade is surprisingly challenging.³⁻⁶ The Handbook on Measuring Digital Trade, a flagship publication prepared jointly by the OECD, WTO, UNCTAD, and the IMF⁴, defines digital trade as all trade that is

digitally ordered and/or deliverable (but not necessarily digitally delivered). That includes, (i) physical trade that is digitally ordered (e.g. purchasing clothes from a foreign online vendor), (ii) trade involving physical services (e.g. using a foreign app to buy a plane ticket), and (iii) trade in digital services that are digitally delivered (e.g. using a foreign file hosting service). The Handbook also “adopts the convention that goods cannot be delivered digitally,” a convention that is at odds with key trade agreements. For instance, the United States–Mexico–Canada Agreement (USMCA) uses the term “digital product” for goods such as a “computer program, text, video, image, sound recording, or other product that is digitally encoded [and] can be transmitted electronically” and the Japan-Switzerland bilateral trade agreement uses the term “digital products” in a definition that includes also digital plans and designs.⁵

These discrepancies are understandable because the distinction between goods and services is not as clear in the digital economy as it is in the physical economy. For instance, entrepreneurs and investors³ often use the term product to indicate service-like activities that are made product-like and scalable through automation and self-service. In that world, people make a strong distinction between the digital delivery of a traditional service (e.g. a remote software engineering team, freelance voice recording) and the digital delivery of a productized service, such as email, maps, or payment platforms. Consider the difference between hiring a human illustrator to generate a drawing and generating one using an AI. The latter, but not the former, scales because it has replaced labor with digital capital in a way that it allows it to service multiple customers at low marginal costs. These productized services, or digital products, are at the core of modern venture capital and include many successful sectors, such as software-as-a-service (e.g. Canva, Photoshop), video streaming (e.g. Netflix, Disney+), and cloud computing (e.g. AWS, Google Cloud).

Our work thus focuses not on all forms of digital trade, but on trade involving digital goods, productized services, and digital intermediation fees (Figure 1), which we call digital products.

³ For example, Marc Andreessen, co-founder of Netscape and the venture capital firm Andreessen Horowitz uses the word products to describe scalable digital goods.⁷ Paul Graham, co-founder of Y-combinator, one of the most influential startup accelerators in Silicon Valley also uses this terminology⁸.

First, we have pure digital goods, such as downloadable video games and movies. Pure digital goods have product like properties, such as a high fixed cost to produce the first copy and a negligible marginal cost to produce additional units (e.g. additional video game downloads). They also involve the transfer of a digital asset, such as a song, movie, or video game. Next, we have productized digital services, which involve access to a digitally encoded and automated service, such as platforms that sell data for a fee, cloud computing, or self-service digital advertising in maps, social media, or search. These productized services range from subscription models that provide access to digital products (e.g. data, movies), to services that run fully online (e.g. advertising on Google or Facebook). Finally, we consider digital transaction fees, but not the physical trade enabled by these platforms. For instance, we consider the fee collected by a travel site selling an airplane ticket, but not the value of the ticket (which involves the flight, a physically delivered service).”

Figure 1. Approximate definition of the bottom-up digital trade data used in this paper. Digital trade is commonly split among digitally and physically delivered trade. In this paper, we adopt a bottom-up definition starting from data on digital firms that includes digital goods, productized services, and transaction fees in digital intermediation platforms

Besides differences in definition, our method also provides some small but important advantages. For instance, our bottom-up approach provides a more granular sectoral classification of digital products than data based on digitally delivered services.

In response to your suggestion, we have also expanded the “Estimating Digital Trade” section. There, we provide again a detailed definition of the digital sectors considered.

“We focus on companies involved in digital goods, productized services, and intermediation (e.g. marketplace platforms). Digital goods, such as video games and software, include products in a digital format with a marginal cost of production that is negligible or close to zero (e.g., eBooks, Software). Productized services, such as cloud computing and video streaming, leverage digital means to automate (almost always fully) the provision of a service. This makes the economics of productized services more similar to those of manufacturing (low marginal cost for each unit and high fixed cost to initiate production). Finally, we consider also fees collected by intermediation platforms, whether these are involved in the purchase of a physically delivered service (e.g. an airplane ticket) or of a digital good or service (e.g. a mobile phone app).”

We also addressed the challenges in classifying digital products. Our categorization of digital products is based on Statista’s Digital Market categories (we have added this to the Estimating Digital Trade section). Acknowledging the reviewer's point, we agree that Online Accommodation and Online Ride-Hailing are distinct categories that should be separated from the Online Marketplaces sector. The revised manuscript now includes these as separate sectors, reflecting a more nuanced understanding of the digital trade landscape. While these additions slightly change our estimates, they do not change our overall findings.

Furthermore, in the “Estimating Digital Trade” section we have included a discussion on how international trade of digital products is currently reported as trade in digitally deliverable services in the Extended Balance of Payments Classification and how our classification aligns with the ISIC conventional classification for economic activities. While there is no direct one-to-

one mapping between our categories and these classifications, we have extended Table S1 to also map our digital categories into the EBOPS and ISIC classifications. This addition provides clarity on how many of our digital sectors align with ISIC's ICT Services section, while also noting that some sectors fall into other sections, such as online accommodation being included in ISIC's 5540 Intermediation service activities for accommodation.

Comment 2

Data and estimation of trade in digital products

Stojkoski et al. take a novel approach to estimating patterns of cross-border trade in digital products, which is welcome and has the potential to offer novel insights. Again, it would be helpful to have a discussion on the relative merits and demerits of their approach compared with that used in other recent studies (e.g. Gonzalez, Sorescu and Kaynak 2023). (Part of the Annex (p23-24) does engage with the UNCTAD 'digitally deliverable services dataset' this is only used as a robustness check).

The authors first estimate the global production of digital products, using corporate revenue data from Orbis and Statista. This occurs in two steps. They use Orbis to identify the 245 largest parent firms (revenues of US\$1bn or more) and their subsidiaries (3,473 firms) that are involved in the trade of digital products and then use Statista data to decompose corporate revenues into 27 different digital product sectors. They also obtain data from a mobile market intelligence company that tracks consumption of applications and games downloaded from Apple's App Store and Google's Play Store. They use this to identify a further 4,812 firms producing digital products in two additional sectors (mobile apps and mobile games). In total, the dataset on production of digital products covers 8,530 firms and 9,446 digital product brands, clustered into 29 distinct digital sectors. The authors are clear on two limits of the dataset: it is biased towards large firms, and it is hard to accurately assign the geographic location of production due to tax optimisation strategies.

To generate an estimate of trade flows, the authors need to obtain country-by-country data on consumption, and here there is very little data. The only data they find is on consumption of mobile apps and games which covers 60 countries. From this data they impute consumption data for a further 69 countries and the remaining 27 product sectors. They predict consumption patterns by introducing a range of country-specific factors (ranging from market size to the level of ICT penetration). Trade estimates are then generated by using an optimal transport approach to assign consumption to the geographically closest company.

The strategy for estimating consumption patterns relies on extrapolating from very few true data points (2 sectors, 60 countries) and while they use impressive methods, the authors should be much more explicit about their assumptions and on the limits of this strategy. What assumptions are being made in extrapolating from consumption patterns in 2 sectors to 29, and how realistic are these assumptions? How plausible are the assumptions underpinning an optimal transport approach in a digital context? For instance, in the case of cloud computing, digital advertising, and online marketplace (the biggest categories in their dataset) are consumption patterns as strongly shaped by geographic factors as in the physical world (e.g. firms tend procure advertising services via their local Google subsidiary / use the closest AWS subsidiary for cloud computing services)? How have they factored differences in the relative importance of geographic distance into their modelling?

***Response:** We thank the reviewer for their thoughtful and in-depth analysis of our methodology. We acknowledge the importance of clearly articulating the assumptions and limitations of our approach.*

To resolve this, we extended both the “Estimating trade in digital products” and the “Discussion” sections. In the “Estimating trade in digital products section” we now include the following paragraph.

“First, the reliance on consumption data primarily from apps and games for forecasting patterns in 29 additional sectors may lead to distortions. This is because the consumption characteristics

of these sectors could differ significantly from those observed in app and mobile games data. Furthermore, our assumption that the international trade patterns of digital products align with geographical proximity, as used in our optimal transport allocation, might not always hold true. While this assumption aligns with standard gravity laws of international trade⁴⁹, the minimal physical constraints in digitally delivered trade might break this law.”

In the “Discussion” section:

“[...] Our estimates are based on several assumptions. One of them was the use of bilateral data on two digital sectors (mobile apps and games) to train our model and to extrapolate our estimates to 29 additional categories. This data limitation might distort the trade patterns of sectors that have significantly different consumption patterns to mobile apps and games which are more consumer oriented, instead of business-to-business oriented sectors (such as cloud computing). In the future, it may be possible to overcome these limitations with the availability of similar bilateral data for other sectors. We also assumed as little trade as possible by maximizing observable domestic consumption. This leads to conservative estimates that can provide only a lower bound for digital product trade volumes. Last, we assumed an optimal transport allocation where trade is assigned to the geographically closest subsidiary. This might not be entirely realistic since digital trade does not involve physical transaction costs.”

Comment 3

Analysis of Trends

Based on these estimates, the authors seek to establish novel insights into the importance and nature of production and consumption of ‘trade in digital products’ relative to other trade flows. To do this, the authors compare their estimates of ‘trade in digital products’ with ‘physical trade’ (their term). They estimate ‘trade in digital products’ to have represented about 5% of world trade in 2021 and expect it to grow to 13% of global trade by 2030. 70% of this trade is from three sectors (cloud computing, online marketplace, and digital advertising) (p.8).

The authors do not engage robustly with recent work on digital trade, which makes it hard to discern the value-add of their analysis. For instance, the WTO estimates that in 2022, ‘digitally delivered services trade’ accounted for 12% of global trade. To what extent is the authors’ analysis of ‘trade in digital products’ already captured conceptually and empirically in existing studies on ‘digitally delivered services’? Alternatively put, how does their estimate of ‘trade in digital products’ add new insights over and above those generated by other studies?

A further problem arises from their omission of services trade when the authors compare ‘trade in digital products’ with wider trade flows, even though services trade accounts for about a quarter of global trade (c.f. Figure 2 which only compares ‘physical trade’ with ‘digital trade’). One of their findings is that trade imbalances in ‘physical trade’ are partially offset when ‘trade in digital products’ is considered, and they cite the example of the US (p. 13). However, this is not a particularly novel insight. It is widely known that while the US has a large trade deficit in goods, this is partially offset by its trade surplus in services. It is widely known that US services trade is increasingly digital, with digitally delivered services accounting for 50% of all US services trade, and 20% of all US exports. The insights into the geographic concentration of ‘trade in digital products’ are fascinating, but again there is recent analysis of the geographic concentration of digital trade flows, which reaches similar conclusions (e.g. Gonzalez, Sorescu and Kaynak 2023 – see p9-10 Annex A).

***Response:** We agree with the reviewer. Comparing our results with known facts on digital trade is indeed an important aspect in order to place our work within the literature. Also, adding total services trade to our analysis can help offset the bias in trade balances that exists in physical goods trade.*

We revised our Results section to always include comparisons with data for trade in digitally deliverable services (and services in general) using WTO-UNCTAD data. We adjust the reported volumes to focus only on digitally delivered services, (using weights provided by EUROSTAT, as given in the Handbook for Measuring Digital Trade).

Our analysis shows that, while both digital products and digitally delivered services (in general) have been growing, digital products have a more rapid expansion rate. This could possibly be because our focus is on the leading companies in the digital sector. We also find a significant relationship between the exports and imports distributions and the bilateral network structure of digital products and digitally delivered services. Finally, our results for trade balances corroborate with trade balances data on digitally delivered services and services in general. That is, most of the economies with digital products positive trade balance, also have positive balance in DDS and overall services trade (since digital products is included within these categories). More importantly, we can use our estimates to investigate the role of digital products trade in the change of total balances. For instance, while the USA has positive balance in digitally delivered services, the positive balance in digital products is even bigger, suggesting that digital products play a large role in decreasing the USA's total trade deficit.

Nevertheless, because we use a narrower, bottom-up definition based on firm revenue data, we end up with lower estimates for the volume of digital trade. This approach allows us to disaggregate bilateral trade flows to a much more granular level (31 sectors compared to a dozen EBOPS categories) and to disentangle the trade structure of a parent firm and its subsidiaries. This, in turn, should aid in tracking statistics that are currently less visible in national accounts (e.g., bilateral trade of Cloud Computing, or evaluating Mode 3: Commercial presence abroad of the General Agreement on Trade in Services), and thus provide a basis for a more detailed data-driven investigations on the implications of digital trade.

Throughout the results section, we then utilize these revised estimates to draw parallels between the growth and geographical spread of digital products trade and digital trade more broadly. Our findings reveal that while both digital products and digitally delivered trade have been experiencing growth over the years, digital products demonstrate a faster growth rate (which could be due to the fact that we focus only on the most important firms in the digital realm). Additionally, we observe a strong correlation between the exports, imports, and network structure of digital products with their digitally delivered services counterparts. This comparative analysis

not only underscores the dynamics of digital product trade but also positions our model as a valuable alternative and complement to the current frameworks used in digital trade research.

These comparisons are now available in three figures on the main text.

Figure 3. Summary statistics and comparisons of trade in digital products. *a* Estimated global trade in digital products in USD (this paper). The error bars show the 95% confidence intervals. *b* Estimated global trade in digitally delivered services in USD (UNCTAD). *c* Global trade in services in USD (UNCTAD) *d* Global trade in physical goods in USD (OEC.world). *e* Estimated composition of trade in digital products compared to services and goods trade in 2021. *f-h* Scatter charts comparing countries' exports in 2021 of digital products to the exports in digitally delivered services (*f*), services (*g*), and goods (*h*) networks. *i-k* Same as *f-h*, just for imports. Figures *f-h* use data only for countries with non-zero digital product exports.

Figure 4. The geography of trade in digital products. *a-d* Spike maps showing the spatial concentration of digital products (*a-b*), digitally delivered services (*c-d*), services (*e-f*), and goods exports and imports in 2021 (*g-h*). *i-h* Lorenz curves for the exports (*i*) and imports (*j*) distributions shown in *a-h*.

Figure 5. The network structure of digital products trade. *a-d* show country-to-country networks of trade in digital products (**a**), digitally delivered services (**b**), services(**c**), and goods (**d**) in 2021. For each country we show the top import and export destination. We also highlight all bilateral trade flows with a volume above USD 1B. *e-g* Scatter charts comparing the countries' eigenvector centrality rank in the digital products trade network to the centralities in the digitally delivered services (**e**), services (**f**), and goods (**g**) networks. **h** Eigenvector centralities for the top 10 countries in terms of eigenvector centralities in all four networks.

Literature referenced by the reviewer:

López González, J., S. Sorescu and P. Kaynak (2023), "Of bytes and trade: Quantifying the impact of digitalisation on trade", OECD Trade Policy Papers, No. 273, OECD Publishing, Paris,<https://doi.org/10.1787/11889f2a-en>.

IMF et al. (2023), Handbook on Measuring Digital Trade, Second Edition, OECD Publishing, Paris/International Monetary Fund, UNCTAD, Geneva 10/WTO, Geneva,<https://doi.org/10.1787/ac99e6d3-en>.

REVIEWER COMMENTS

Reviewer #2 (Remarks to the Author):

The paper was already a strong and innovative contribution to the literature and it has now been much improved.

- There is now a very clear link between the literature on measuring digital trade and the contribution made by this paper. The comparisons across different categories of trade (goods, services, DDS) are also much appreciated.
- Particularly neat are the findings about spatial concentration. These give new evidence on the high concentration of supply (80% of digital products originates in the top 3% of countries) and the low concentration, relative to other services, of demand (80% of digital product imports go to 22% of countries).
- The finding regarding the concentration of digital product networks around the US (and how this is much more pronounced than what we see in goods) is also important. While it is something that has been well accepted, it has not been well documented empirically (indeed, recent analysis shows that the share of US in digital trade might be falling). Also surprising is that China does not seem to be in the picture.

Overall, I believe this is a great contribution to the emerging literature on digital trade.

One last issue, which I do not think should hold publication, relates to trade balances. I appreciate the efforts made in this section. Your exposition is now very clear, and shows goods and services balances as well as digital product balances, giving new insights into what these tell you and how they don't always go in the same direction (I liked the use of quadrants to highlight this). I'm still struggling to understand whether or not you might be capturing something that is unrecorded in trade statistics. For example, services trade statistics are not well suited to capture mode 3 (commercial presence abroad). However, it occurs to me that your measure of digital products might actually capture mode 3 services. This is because it is consumption based and attributes value across foreign affiliates. If that is the case, then this is a valuable contribution to the literature and offers new and important insights re. trade balances. However, if this is not the case, then we don't learn much new from your claim that the goods trade balance of the US is reduced by 24% when you account for digital products.

A few small comments to note.

- Lines 39-40. The Handbook defines digital trade as "All international trade that is digitally ordered and/or digitally delivered" (see page 22 of the Handbook). You claim that it is "digitally deliverable" but this is somewhat of a judgement call regarding how this is being measured currently.
- Lines 43-49. The convention of defining digital deliveries as services, while at odds with definitions in some trade agreements is perfectly aligned with existing statistical approaches. Indeed, the UN International Merchandise Trade Manual, The Balance of Payments Statistics Manual (BPM6) and The Central Product Classification all recognise that transactions involving content challenge distinctions between goods and services, but they converge in suggesting that these should be recorded in the services accounts (might be worth adding this nuance).
- Lines 51-62. There are ongoing and longstanding discussions in policy circles about whether these digital goods/digital products/electronic transmissions should be treated as goods or as services. This is largely in the context of the WTO Moratorium on applying customs duties on electronic transmissions (See here for a recent discussion of scope, definition and impact issues). It might be useful to add a short reference to the discussions. Note that if the Moratorium falls, some of the transactions you are capturing in your digital product statistics might be subject to tariffs!
- L 109-113. When the trade balance discussion pits merchandise trade against estimates of digital products. I don't see how comparing goods balance with digital products balance is relevant unless you can claim that you are capturing something that is not being captured in existing trade statistics (mode 3). The goods trade balance can be offset by the services trade balance, but selecting one

category of 'services' as offsetting is somewhat strange.

- L 184. You make the argument later in the discussion (L 526-527) but when I was reading this I thought immediately of B2B, thought useful to flag here.
- When calculating economic complexity, your comparator appears to be goods trade. Would not a better comparator be DDS or services?

Reviewer #3 (Remarks to the Author):

The authors have addressed the points that were raised in the review. The revised version is substantially improved. It engages with the existing literature, the shortcomings of the measurement techniques are made clearer, and in general, I find the analysis more robust and persuasive.

The one section where I still have substantial concerns is in the analysis and discussion of trade balances (page 18 and figures 6a and 6b). In the response to reviewer 1, the authors state "We now compare total trade balances (encompassing both services and goods) with and without the inclusion of digital products (Figure 6 a and b). This approach allows us to assess the impact of digital product trade more accurately on a country's overall trade balance. We present this analysis through plots that compare the total trade balance against the adjusted trade balance, thereby illustrating the impact of digital product trade on national economies." While this approach makes sense as it would help us understand the role of digital products in the overall trade balance, it is not apparent in the main text that this is actually what the authors are doing. Figure 6a "compares per capita trade balances for digital products and trade in goods and services". If this is the case, why is this comparison being made? What new insights do we derive from knowing which quadrant a country falls into?

Similarly, in Figure 6b, it is interesting to see differences in the trade balances between digital products and digitally delivered services (e.g. UK vs Ireland). This provides some granularity of insight that is missing from more aggregated data on digital trade. However, it is unclear to me what value we gain from comparing these balances in digital trade to the overall trade in goods balance (e.g. the statement that the US goods trade deficit is reduced by 24% when digital products are included). Why is this relevant and useful? Related to this, the discussion section states that "we found that while trade in digital products represents a relatively small fraction of the global economy, it can significantly change estimates of trade balances for net digital product exporters and importers". It's not clear from the analysis of trade balances that this is a general finding – i.e. one that holds beyond the very few countries that are major exporters of digital products.

Response to Reviewers

In the following pages we provide detailed answers to the reviewers' comments (in *italics and in blue*). We have also highlighted changes to the manuscript.

Reviewer #2 (Remarks to the Author):

Overall Assessment

The paper was already a strong and innovative contribution to the literature and it has now been much improved.

- There is now a very clear link between the literature on measuring digital trade and the contribution made by this paper. The comparisons across different categories of trade (goods, services, DDS) are also much appreciated.
- Particularly neat are the findings about spatial concentration. These give new evidence on the high concentration of supply (80% of digital products originates in the top 3% of countries) and the low concentration, relative to other services, of demand (80% of digital product imports go to 22% of countries).
- The finding regarding the concentration of digital product networks around the US (and how this is much more pronounced than what we see in goods) is also important. While it is something that has been well accepted, it has not been well documented empirically (indeed, recent analysis shows that the share of US in digital trade might be falling). Also surprising is that China does not seem to be in the picture.

Overall, I believe this is a great contribution to the emerging literature on digital trade.

Response: We thank the reviewer for their insightful and encouraging feedback, and for recognizing the improvements made in our revised manuscript.

Comment 1

One last issue, which I do not think should hold publication, relates to trade balances. I appreciate the efforts made in this section. Your exposition is now very clear, and shows goods and services balances as well as digital product balances, giving new insights into what these tell you and how they don't always go in the same direction (I liked the use of quadrants to highlight this). I'm still struggling to understand whether or not you might be capturing something that is unrecorded in trade statistics. For example, services trade statistics are not well suited to capture mode 3 (commercial presence abroad). However, it occurs to me that your measure of digital products might actually capture mode 3 services. This is because it is consumption based and attributes value across foreign affiliates. If that is the case, then this is a valuable contribution to the literature and offers new and important insights re. trade balances. However, if this is not the case, then we don't learn much new from your claim that the goods trade balance of the US is reduced by 24% when you account for digital products.

Response: We agree with the reviewer that we should clarify how our methodology refines existing trade data. We also acknowledge and agree the observation that our approach, through its headquarters-based assignment, could indeed shed light on GATS Mode 3: Commercial Presence; a mode that is difficult to capture using service trade statistics. To add clarity to the paper we have added a new figure comparing goods and service trade balances with the digital product trade balances obtained using the headquarters assignment. This figure is expected to capture GATS mode 3 and shows, for instance, that the United States enjoys a large and positive digital product trade balance (>1k USD per capita) when its goods and services trade balance is negative. Conversely, tax havens, such as Luxembourg, which enjoys a positive digital product trade balance based when using subsidiary assignments, has a large and negative trade balance when we use headquarters assignment (~ 3k USD per capita). We also added some text to the paper discussing GATS modes of trade. We hope these two changes help incorporate the reviewers' comments. Once again, we thank the reviewer for helping us improve the clarity of our contribution.

Figure 6. Implications of trade in digital products. *a* Total trade balance (goods + services) vs digital product trade balance (USD per capita) in 2021 using the subsidiary assignment for digital products trade. *b* Same as *a*, only using the headquarters assignment for digital products trade. *c* Average digital product, DDS, services, and goods exports per capita between 2016 and 2021 for high income economies depending on whether they decoupled growth from emissions or not (DE – decoupled, Non-DE – not decoupled). We highlight the regions enclosed by the 25th and 75th percentiles. *d* Change in economic complexity index estimates after incorporating trade in digital products to data on physical trade. Inset shows boxplots for the PCI of digital products and physical products in 2021.

Minor comments:

A few small comments to note.

- Lines 39-40. The Handbook defines digital trade as “All international trade that is digitally ordered and/or digitally delivered” (see page 22 of the Handbook). You claim that it is “digitally deliverable” but this is somewhat of a judgement call regarding how this is being measured currently.

Response: We thank the reviewer for this comment. The revised sentence now reads:

“The Handbook on Measuring Digital Trade, a flagship publication prepared jointly by the OECD, WTO, UNCTAD, and the IMF4, defines digital trade as all trade that is digitally ordered and/or delivered.”

- Lines 43-49. The convention of defining digital deliveries as services, while at odds with definitions in some trade agreements is perfectly aligned with existing statistical approaches. Indeed, the UN International Merchandise Trade Manual, The Balance of Payments Statistics Manual (BPM6) and The Central Product Classification all recognise that transactions involving content challenge distinctions between goods and services, but they converge in suggesting that these should be recorded in the services accounts (might be worth adding this nuance).

Response: We again thank the reviewer for their comment. We added this as a footnote to the sentence. The footnote reads:

“Nevertheless, this definition aligns with current statistical approaches. The UN International Merchandise Trade Manual, The Balance of Payments Statistics Manual (BPM6), and The Central Product Classification all recognize that transactions involving content challenge distinctions between goods and services, but they still suggest recording them in the services accounts.”

- Lines 51-62. There are ongoing and longstanding discussions in policy circles about whether these digital goods/digital products/electronic transmissions should be treated as goods or as services. This is largely in the context of the WTO Moratorium on applying customs duties on electronic transmissions (See here for a recent discussion of scope, definition and impact issues). It might be useful to add a short reference to the discussions. Note that if the Moratorium falls, some of the transactions you are capturing in your digital product statistics might be subject to tariffs!

***Response:** We thank the reviewer for pointing this out. We have added a reference and a mention to the discussions in the paper. In particular, at the end of the paragraph we added the following sentences:*

“Indeed, recently a critical policy discussion has emerged regarding the classification of such digital products under trade agreements.⁹⁻¹¹ The outcome of these policy discussions could have a profound impact on the sector, potentially subjecting some digital product transactions to tariffs, thereby affecting the statistics capturing the economic contributions of these digital products.”

- L 109-113. When the trade balance discussion pits merchandise trade against estimates of digital products. I don't see how comparing goods balance with digital products balance is relevant unless you can claim that you are capturing something that is not being captured in existing trade statistics (mode 3). The goods trade balance can be offset by the services trade balance, but selecting one category of 'services' as offsetting is somewhat strange.

***Response:** We agree with the reviewer that our GATS modes of trade definition is unclear, since our headquarters assignment captures trade corresponding to mode 3. Following the reviewer comments we have brought the headquarters assignment into the paper by adding a new figure (6B). We agree that this is a more interesting/relevant comparison, since mode 3 is not well captured by trade statistics.*

- L 184. You make the argument later in the discussion (L 526-527) but when I was reading this I thought immediately of B2B, thought useful to flag here.

Response: We agree with the reviewer that mentioning this caveat is important. We include a comment on this later in the same section, when discussing the caveats of all our datasets.

- When calculating economic complexity, your comparator appears to be goods trade. Would not a better comparator be DDS or services?

Response: This is an important question. We have added a footnote in the paper where we introduce the HS classification goods data (used to calculate the economic complexity index) to clarify in case our readers have a similar concern. In brief, economic complexity calculations require highly disaggregate data which is available for the trade of goods and not for the trade of services. Goods trade is usually categorized using the four or six digit harmonized system which involve, respectively, 1,000+ or 5,000+ unique categories. Service trade is much more aggregate (a dozen categories), and hence, cannot be used to estimate economic complexity.

Reviewer #3 (Remarks to the Author):

Overall Assessment

The authors have addressed the points that were raised in the review. The revised version is substantially improved. It engages with the existing literature, the shortcomings of the measurement techniques are made clearer, and in general, I find the analysis more robust and persuasive.

Response: We thank the reviewer for valuing our revision and for helping us improve the quality of our contribution.

Comment 1

The one section where I still have substantial concerns is in the analysis and discussion of trade balances (page 18 and figures 6a and 6b). In the response to reviewer 1, the authors state “We now compare total trade balances (encompassing both services and goods) with and without the inclusion of digital products (Figure 6 a and b). This approach allows us to assess the impact of digital product trade more accurately on a country's overall trade balance. We present this analysis through plots that compare the total trade balance against the adjusted trade balance, thereby illustrating the impact of digital product trade on national economies.” While this approach makes sense as it would help us understand the role of digital products in the overall trade balance, it is not apparent in the main text that this is actually what the authors are doing. Figure 6a “compares per capita trade balances for digital products and trade in goods and services”. If this is the case, why is this comparison being made? What new insights do we derive from knowing which quadrant a country falls into?

Similarly, in Figure 6b, it is interesting to see differences in the trade balances between digital products and digitally delivered services (e.g. UK vs Ireland). This provides some granularity of insight that is missing from more aggregated data on digital trade. However, it is unclear to me what value we gain from comparing these balances in digital trade to the overall trade in goods balance (e.g. the statement that the US goods trade deficit is reduced by 24% when digital products are included). Why is this relevant and useful? Related to this, the discussion section states that “we found that while trade in digital products represents a relatively small fraction of the global economy, it can significantly change estimates of trade balances for net digital product exporters and importers”. It’s not clear from the analysis of trade balances that this is a general finding – i.e. one that holds beyond the very few countries that are major exporters of digital products.

Response: We agree with the reviewer that our exposition of trade balances and their relevance was unclear. In particular, what made our estimates interesting is that when we use the headquarters assignment rule, they capture some of GATS mode 3 trade (commercial presence),

which has been traditionally hard to capture in trade data. We have updated the manuscript to include the GATS modes definitions and have changed figure 6B to focus on the headquarter assignment trade. We have also updated the text to clarify why this mode of trade is relevant. We thank the reviewer for the comment which has improved the quality of our contribution.

Figure 6. Implications of trade in digital products. *a* Total trade balance (goods + services) vs digital product trade balance (USD per capita) in 2021 using the subsidiary assignment for digital products trade. *b* Same as *a*, just using the headquarter assignment for digital products trade. *c* Average digital product, DDS, services, and goods exports per capita between 2016 and 2021 for high income economies depending on whether they decoupled growth from emissions or not (DE – decoupled, Non-DE – not decoupled). We highlight the regions enclosed by the 25th and 75th percentiles. *d* Change in economic complexity index estimates after incorporating trade in digital products to data on physical trade. Inset shows boxplots for the PCI of digital products and physical products in 2021.

REVIEWERS' COMMENTS

Reviewer #3 (Remarks to the Author):

Congratulations! In my opinion, this is now a really strong piece of work and it makes an important contribution to the literature.